# Fasting alters the gut microbiome reducing blood pressure and body weight in metabolic syndrome patients

András Maifeld [1,2,3,4], Hendrik Bartolomaeus [1,2,3,4], Ulrike Löber[1,3,4], Ellen G. Avery [1,3,4,5], Nico Steckhan [2,6], Lajos Markó [1,2,3], Nicola Wilck [1,3,7,8], Ibrahim Hamad[9,10], Urša Šušnjar[1], Anja Mähler[1,2,3], Christoph Hohmann[6], Chia-Yu Chen[1,2,3,4], Holger Cramer[11], Gustav Dobos[11], Till Robin Lesker[12], Till Strowig [12,13], Ralf Dechend[1,2,3,14], Danilo Bzdok [15,16,17], Markus Kleinewietfeld [9,10], Andreas Michalsen [2,6,18✉], Dominik N. Müller [1,2,3,4,18✉] & Sofia K. Forslund [1,2,3,4,18✉]

Periods of fasting and refeeding may reduce cardiometabolic risk elevated by Western diet. Here we show in the substudy of NCT02099968, investigating the clinical parameters, the immunome and gut microbiome exploratory endpoints, that in hypertensive metabolic syndrome patients, a 5-day fast followed by a modified Dietary Approach to Stop Hypertension diet reduces systolic blood pressure, need for antihypertensive medications, body-mass index at three months post intervention compared to a modified Dietary Approach to Stop Hypertension diet alone. Fasting alters the gut microbiome, impacting bacterial taxa and gene modules associated with short-chain fatty acid production. Cross-system analyses reveal a positive correlation of circulating mucosa-associated invariant T cells, non-classical monocytes and CD4+ effector T cells with systolic blood pressure. Furthermore, regulatory T cells positively correlate with body-mass index and weight. Machine learning analysis of baseline immunome or microbiome data predicts sustained systolic blood pressure response within the fasting group, identifying CD8+ effector T cells, Th17 cells and regulatory T cells or Desulfovibrionaceae, Hydrogenoanaerobacterium, *Akkermansia*, and Ruminococcaceae as important contributors to the model. Here we report that the high-resolution multi-omics data highlight fasting as a promising non-pharmacological intervention for the treatment of high blood pressure in metabolic syndrome patients.

A full list of author affiliations appears at the end of the paper.

Fasting can prolong survival and reduce disease burden in rodent models, and possibly in humans[1]. In contrast, today's Western diet promotes cardiometabolic disease (CMD)[2]. How diet affects the gut microbiota, immune system and subsequently host (patho)physiology is not fully understood, and information is lacking on how periodic fasting affects the gut microbiome in patients with metabolic syndrome (MetS). To reduce CMD risk, exercise and a healthy diet are often prescribed. Shifting from a "Western diet" to a healthier "Mediterranean-like" DASH diet[3] to achieve optimal nutrition and negative energy balance is recommended, although compliance is a major hurdle. Our study is the first of its kind to investigate the effects of a lifestyle modification in combination with fasting therapy in patients with MetS using a multi-omics approach by combining gut microbiome analysis and deep immunophenotyping. The "Western diet" is known to induce metabolic inflammation, accelerating CMD[4]. The gut microbiota is a delicate ecosystem that plays a pivotal role in health and disease. Dysbiosis has been observed as a characteristic of several inflammatory, cardiovascular, and metabolic disorders (e.g. obesity)[5], including hypertension[6,7]. The "healthy" gut microbiome is relatively stable, although various factors such as antibiotics, intestinal infections, and profound dietary or lifestyle changes, such as moving on or off a "Western diet", can induce transient or persistent changes to this ecosystem. Traditionally, fasting plays an important role in different cultural and religious practices. Dramatic caloric restriction not only affects host health and physiology, but also has an impact on the microbiome[8–10]. Here, we studied the role of fasting in cardiovascular risk patients with MetS (Table 1). Five days of fasting followed by 3 months of a modified DASH diet induced distinct microbiome and immunome changes not seen under DASH alone, as well as a sustained SBP benefit even 3 months post-intervention. Applying machine-learning algorithms, we were able to make effective predictions regarding which patients would respond positively to treatment via BP reduction from either baseline immunome or 16S microbiome data. The microbial signature for BP responsiveness generalizes to a recently published cohort investigating the impact of fasting in 15 healthy male volunteers, as do many of the microbiome changes upon fasting. These data highlight fasting followed by a shift to a health-promoting diet as a promising non-pharmacological intervention for patients with hypertensive MetS, with possible implications for a wider spectrum of health states.

## Results

**Fasting affects the gut microbiome and immunome.** As we have previously reported a major influence of common MetS drugs on the microbiota[11], we accounted for any changes in medication regime or dosage in our statistical tests, alongside controlling for important demographic features such as age and sex. There were substantial and significant (PERMANOVA $P = 0.001$) differences in microbial composition within individuals during fasting, reflecting a characteristic intervention-induced shift, which later partially reverted following a 3-month refeeding period on a DASH diet (Fig. 1d, Supplementary Data 1 and Fig. 1a). This was echoed by analogous significant (PERMANOVA $P = 0.001$) changes in host immune cell composition during the intervention, revealing a fasting-specific signature, which likewise largely reversed during refeeding (Fig. 1e, Supplementary Data 1). We did not observe significant changes to the microbiome species richness/alpha diversity (between-group Mann–Whitney U (MWU) $P > 0.05$, within-individual likelihood ratio test FDR > 0.1 for all comparisons; Supplementary Data 2; Shannon: Fig. 1b, Supplementary Fig. 2) after either fasting or refeeding in the

**Table 1 Patient characteristics at baseline.**

|  | FASTING + DASH | DASH |
|---|---|---|
| Females/Males | 23/12 | 21/15 |
| Age (year) | 58 ± 8 | 62 ± 8 |
| Height (cm) | 171 ± 8 | 171 ± 9 |
| Office SBP (mm Hg) | 136 ± 15 | 138 ± 16 |
| Office DBP (mm Hg) | 88 ± 11 | 88 ± 9 |
| 24 h ABPM SBP (mm Hg) | 132 ± 9 | 131 ± 9 |
| 24 h ABPM DBP (mm Hg) | 81 ± 8 | 81.4 ± 7 |
| 24 h ABPM MAP (mm Hg) | 104 ± 8 | 104 ± 7 |
| 24 h ABPM peripheral resistance (mm Hg*s/ml) | 1.4 ± 0.1 | 1.3 ± 0.1 |
| SBP day (mm Hg) | 134 ± 10 | 133 ± 10 |
| DBP day (mm Hg) | 83 ± 9 | 84 ± 7 |
| SBP nocturnal (mm Hg) | 120 ± 12 | 121 ± 10 |
| DBP nocturnal (mm Hg) | 71.5 ± 8 | 71.6 ± 7 |
| Weight (kg) | 99 ± 17 | 96 ± 17 |
| BMI (kg/m$^2$) | 34 ± 4.9 | 33 ± 4.7 |
| Hip circumference (cm) | 115 ± 20 | 113 ± 17 |
| Waist circumference (cm) | 116 ± 11 | 114 ± 12 |
| Waist to hip ratio | 1.1 ± 0.7 | 1.0 ± 0.2 |
| Body fat percentage (%) | 42 ± 8 | 39 ± 10 |
| HOMA index | 2.8 ± 2.1 | 3.4 ± 2.4 |
| Insulin (mU/l) | 10.4 ± 6.4 | 12.1 ± 7.4 |
| Plasma glucose (mg/dl) | 105 ± 20 | 110 ± 20 |
| Hb-A1C (%) | 5.8 ± 0.4 | 5.9 ± 0.7 |
| Hb-A1C IFCC (mmol/mol) | 39.6 ± 4.8 | 41.2 ± 7.4 |
| Triglyceride (mg/dl) | 166 ± 106 | 169 ± 109 |
| Cholesterol (mg/dl) | 220 ± 48 | 222 ± 54 |
| HDL (mg/dl) | 50 ± 11 | 51 ± 10 |
| LDL(mg/dl) | 137 ± 36 | 140 ± 45 |
| LDL/HDL ratio | 2.8 ± 0.7 | 2.8 ± 0.9 |
| CRP (mg/l) | 0.4 ± 0.4 | 0.3 ± 0.3 |
| IL-6 (pg/ml) | 3.1 ± 2.0 | 2.8 ± 2.2 |
| Creatinine (mg/dl) | 0.9 ± 0.2 | 0.9 ± 0.2 |
| eGFR Cockroft-Gault (ml/min) | 120 ± 39 | 107 ± 32 |

Mean values and $+/-$ one standard deviation are shown

present dataset, though a trend of reduced, then restored diversity was seen in the longitudinal tests. Similarly, there were no significant changes between time points in the intersample gut taxonomic variability/beta diversity (Bray–Curtis distance, Fig. 1c. DASH without fasting neither affected the microbial composition nor the host immune cell composition ($P = 0.374$ and $P = 0.378$, respectively, Supplementary Fig. 1B, C).

Fasting resulted in a reduction of CD3$^+$, CD4$^+$ T cells, and CD19$^+$ B cells, while the frequency of CD8$^+$ T cells was unaltered. In contrast, fasting increased the abundance of monocytes (CD14$^+$CD11c$^+$CD19$^-$CD3$^-$) and TCRγ/δ$^+$ T cells. However, these changes were reversed upon refeeding (Fig. 1h, Supplementary Data 1). Of note, frequency of CD123$^+$CD14$^-$CD16$^-$HLA-DR$^+$ plasmacytoid dendritic cells also increased upon fasting and was still enriched after refeeding (Fig. 1h, Supplementary Data 1). When looking closer into monocyte subsets, fasting increased (and refeeding reduced) the frequency of classical CD14$^{high}$CD16$^-$, non-classical CD14$^{low}$CD16$^{++}$, and intermediate CD14$^{high}$CD16$^+$ monocytes (Fig. 1i, Supplementary Fig. 1, Data 1), which was confirmed by unbiased FlowSOM analyses (Supplementary Fig. 4A–D). Fasting also affected the relative abundance of differentially activated T cells. Upon fasting, CD8$^+$ T cells showed a higher percentage of terminally differentiated cells (T$_{eff}$ CD45RO$^-$CD62L$^-$) and a lower percentage of the naïve phenotype (T$_n$, CD45RO$^-$CD62L$^+$), while memory T cells were not affected (Fig. 1i, Supplementary Fig. 3, Data 1). A similar

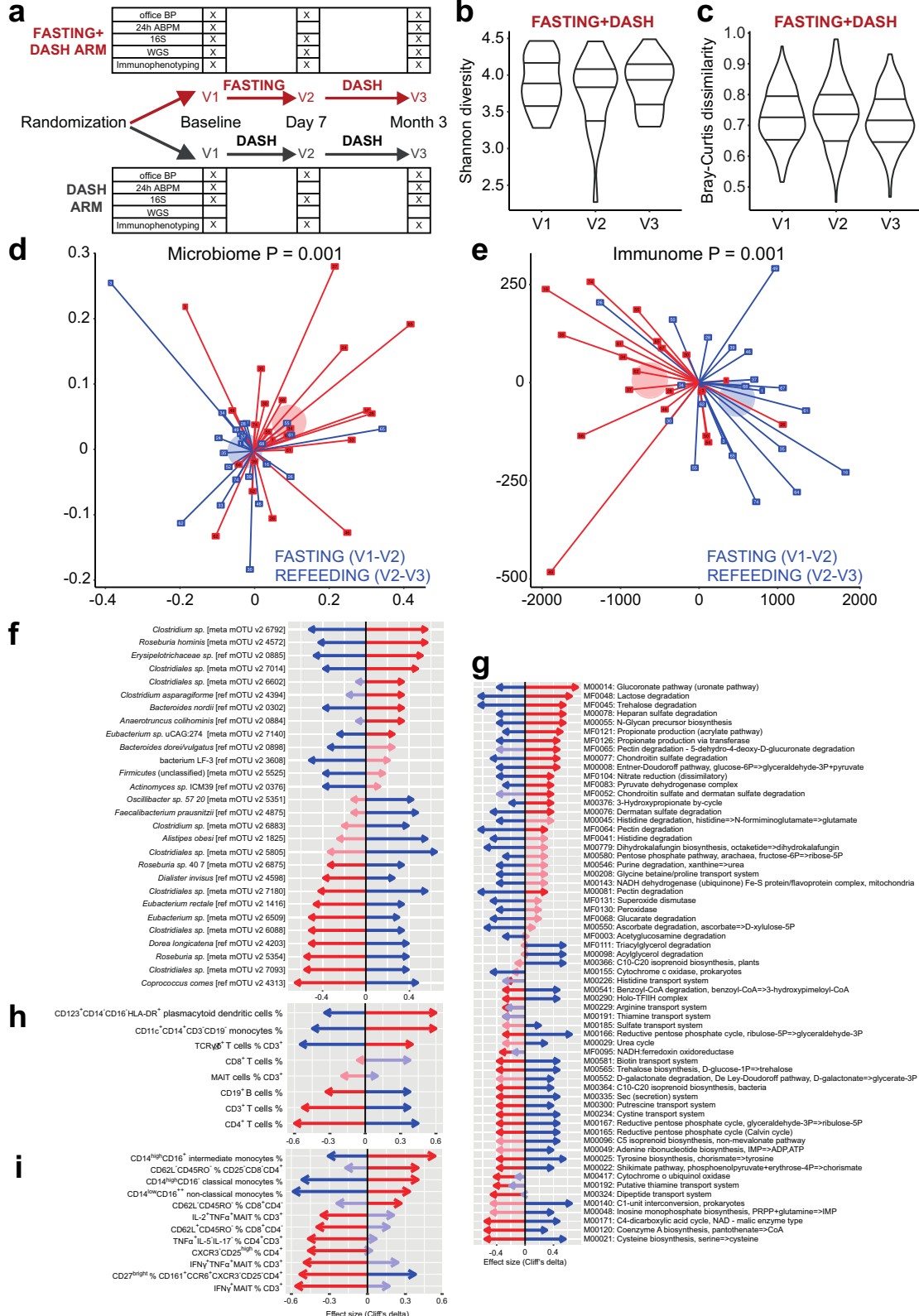

pattern was observed in CD4$^+$ T$_{eff}$ (Fig. 1i, Supplementary Data 1). Further, fasting decreased the frequency of pro-inflammatory Th17 (CD27$^{bright}$ CD161$^+$CCR6$^+$CXCR3$^-$CD25$^-$CD4$^+$), as well as TNFα- and IFNγ-producing Th1 cells (Fig. 1i, Supplementary Data 1). These changes were partially reverted upon refeeding (Fig. 1i). Neither fasting nor refeeding changed the overall frequency of CD161$^+$Vα7.2$^+$ CD3$^+$ mucosa-

associated invariant cells (MAIT, Fig. 1h, Supplementary Fig. 3). However, frequency of pro-inflammatory MAITs producing TNFα and IFNγ significantly decreased upon fasting and were minimally affected by refeeding (Fig. 1i, Supplementary Data 1).

Next, we tested all gut microbial taxa and gene functional (KEGG[12], GMM[13]) modules for abundance shifts during fasting or refeeding, as well as persistent shifts across the 3-month study

**Fig. 1 Fasting has a pervasive host and microbiome impact. a** Study design is shown. Subjects are followed from baseline (V1), randomly assigned to begin a modified DASH diet only or to undergo a 5-day fast followed by a modified DASH diet. Follow-up is done at one week (V2) and 3 months (V3). **b** Fasting has no significant (two-sided MWU $P > 0.05$) impact on gut microbiome alpha diversity (Shannon diversity from mOTUv2 OTUs) across observation times V1–V3. **c** Fasting has no significant (two-sided MWU $P > 0.05$) impact on gut microbiome beta diversity (Bray–Curtis dissimilarity from mOTUv2 OTUs, shown are all between donor comparisons per time point) across observation times V1–V3. **d** Fasting significantly shifts the gut microbiome towards a characteristic compositional state, while refeeding reverses this change. Unconstrained Principal Coordinates graph with first two dimensions shown. Axes show Bray–Curtis dissimilarities of rarefied mOTUv2 OTUs between samples; each participant in the fasting arm is shown as two lines, one red (fasting change), one blue (refeeding change) connected (centered) at the origin for ease of visualization. Axes show fasting and refeeding deltas after one-week intervention and 3-month refeeding. Pseudonym participant ID numbers are shown on the point markers. Transparent circle markers show arithmetic mean position of fasting and recovery deltas, respectively. PERMANOVA test $P$-values reveal significant dissimilarity ($P < 0.05$) between samples from each visit V1–V3 in the original distance space, stratifying by donor. **e** Fasting significantly shifts the host immune cell population towards a characteristic state, while refeeding reverses it. Same as in (**d**), using Euclidean distances. **f** Gut microbial taxa significantly enriched/depleted upon fasting/refeeding. Taxa (mOTUv2 OTUs) are shown on the vertical axis, and effect sizes (Cliff's delta) shown on the horizontal axis. Red arrows represent fasting effects (V2–V1 comparison), blue arrows refeeding effects (V3–V2 comparison). Bold arrows are significant (nested model comparison of a linear model for rarefied abundance of each taxon, comparing a model incorporating patient ID, age, sex and all dosages of relevant medications) to a model additionally incorporating time point, requiring likelihood test Benjamini-Hochberg corrected FDR < 0.1 and additionally pairwise post-hoc two-sided MWU test $P < 0.05$. **g** Gut microbial gene functional modules (KEGG and GMM models analyzed together) significantly enriched/depleted upon fasting/refeeding. **h** General immune cell populations significantly enriched/depleted upon fasting/refeeding. **i** Specific immune cell subpopulations. **g–i** Same test as in (**f**), subset of altered features shown for clarity. Effect sizes and FDR-corrected $P$ values can be found in Supplementary Data 1,2.

period, controlling for age, sex and any changes in medication (Fig. 1f–h, Supplementary Data 1). Fasting stimulated shifts in the abundance of several core commensals, which were reversed upon refeeding (Fig. 1f, Supplementary Data 1). Many Clostridial Firmicutes shifted significantly in abundance, with an initial decrease in butyrate producers such as *Faecalibacterium prausnitzii*, *Eubacterium rectale* and *Coprococcus comes*, which had also reverted after 3 months. Interestingly, modeling the shift in *C. comes* abundance as a function of body-mass index (BMI) changes during the study yielded a better fit of the data than when it was modeled as a function of the fasting intervention. Bacteroidaceae showed the opposite pattern. At the end of the refeeding period, a persistent depletion could be seen in Enterobacteriaceae, especially *Escherichia coli*. These shifts were accompanied by vast changes in microbial metabolic capacity (Fig. 1g, Supplementary Data 1). Fasting enriched for propionate production capacity, mucin degradation gene modules, and diverse nutrient utilization pathways.

Reanalyzing previously published data, we compared the microbiome signatures of metformin use and MetS to those seen in our dataset[11,14]. For ease of comparability, we proceeded with only human gut-specific functional modules (GMM) assessed from shotgun sequencing data available for the fasting arm. Certain fasting- or refeeding-associated functional gene modules from our data were found to overlap with signatures of metformin usage or MetS, though there was little concordance on a taxonomic composition level, in line with previously described higher functional than taxonomic concordance between microbiomes. Of note, when comparing the metformin signal to the MetS signal, it is clear that these two effects are functionally distinct and often oppose one another. In contrast, the inferred gut functional signature of metformin treatment shared some features with that of our fasting intervention (Supplementary Fig. 5).

**Fasting reduces long-term systolic blood pressure and body weight in MetS patients**. Assessing the clinical relevance of our intervention, we inspected clinical outcomes in the two study arms. While DASH reduced office SBP after 3 months (Fig. 2h), it did not significantly (MWU $P = 0.27$) affect 24 h ambulatory SBP, the gold standard of clinical BP measurements (Fig. 2a)[3]. In contrast, fasting followed by a modified DASH diet led to a sustained reduction both in 24 h ambulatory SBP and mean arterial pressure (MAP) (MWU $P < 0.05$, Fig. 2a). Further,

subjects undergoing fasting could significantly ($\chi^2$ $P = 0.035$) reduce their intake of antihypertensive medication in 43% of cases, compared to only 17% of the cases on DASH alone, while their BP remained under control (Fig. 2b, Supplementary Data 3). Because the BP response to fasting was heterogeneous in our cohort (Fig. 2a, b), we applied a decision tree model to stratify patients based on their ambulatory BP response, adjusted for antihypertensive medication (Supplementary Fig. 6, Data 4). The responder group ($n = 22$) had a median SBP decrease of 8.0 mmHg, irrespective of the high reduction in medications amongst these patients, while the decrease in the non-responder group ($n = 10$) was significantly lower (0.3 mmHg; Fig. 2c). In the DASH-only arm, 17 patients were classified as responders with a median SBP decrease of 8.0 mmHg, while the non-responders ($n = 14$) showed no decrease in median SBP (0.5 mmHg, Fig. 2c). Fasting followed by a modified DASH diet, unlike a modified DASH diet alone, significantly (drug-adjusted post-hoc $P < 0.05$) reduced BMI and body weight even 3 months post-fasting (Fig. 2d, e). Although all fasting+DASH participants showed a reduction in body weight, this reduction alone could not explain the long-term ambulatory SBP and MAP changes exclusive to the fasting arm (Fig. 2f, g), nor the microbiome or immunome changes accompanying it. 95% of significant findings retain significance when BMI is added as a predictor to the nested models for longitudinal data (see Supplementary Data 5). Very few of the significant effects observed in the fasting+DASH arm could be replicated in the equally powered DASH-only arm (Fig. 3a–c).

**BP responder-specific changes in the gut microbiome and immunome**. Because the BP responsiveness was heterogeneous in the fasting+DASH arm (Fig. 2a–c), despite the similar disease severity indicated by the baseline clinical characteristics of these patients (Supplementary Data 6), we hypothesized that unique characteristics involving the immunome or microbiome of these patients may contribute to their BP response. We compared the impact of fasting and refeeding in the complete fasting arm, in the BP responders of the fasting arm, and in the DASH-only arm (Fig. 4a, b, Supplementary Data 2, 7, 8). Even at reduced statistical power, we were able to capture changes in the abundance of many gut microbial taxa that were uniquely characteristic of successful fasting treatment even 3 months post-fasting (Fig. 4a, Supplementary Data 7, 8). Fasting combined with DASH resulted in the sustained depletion of Actinobacteria family members Corynebacteriaceae and Actinomycetaceae (Fig. 4a). BP responders

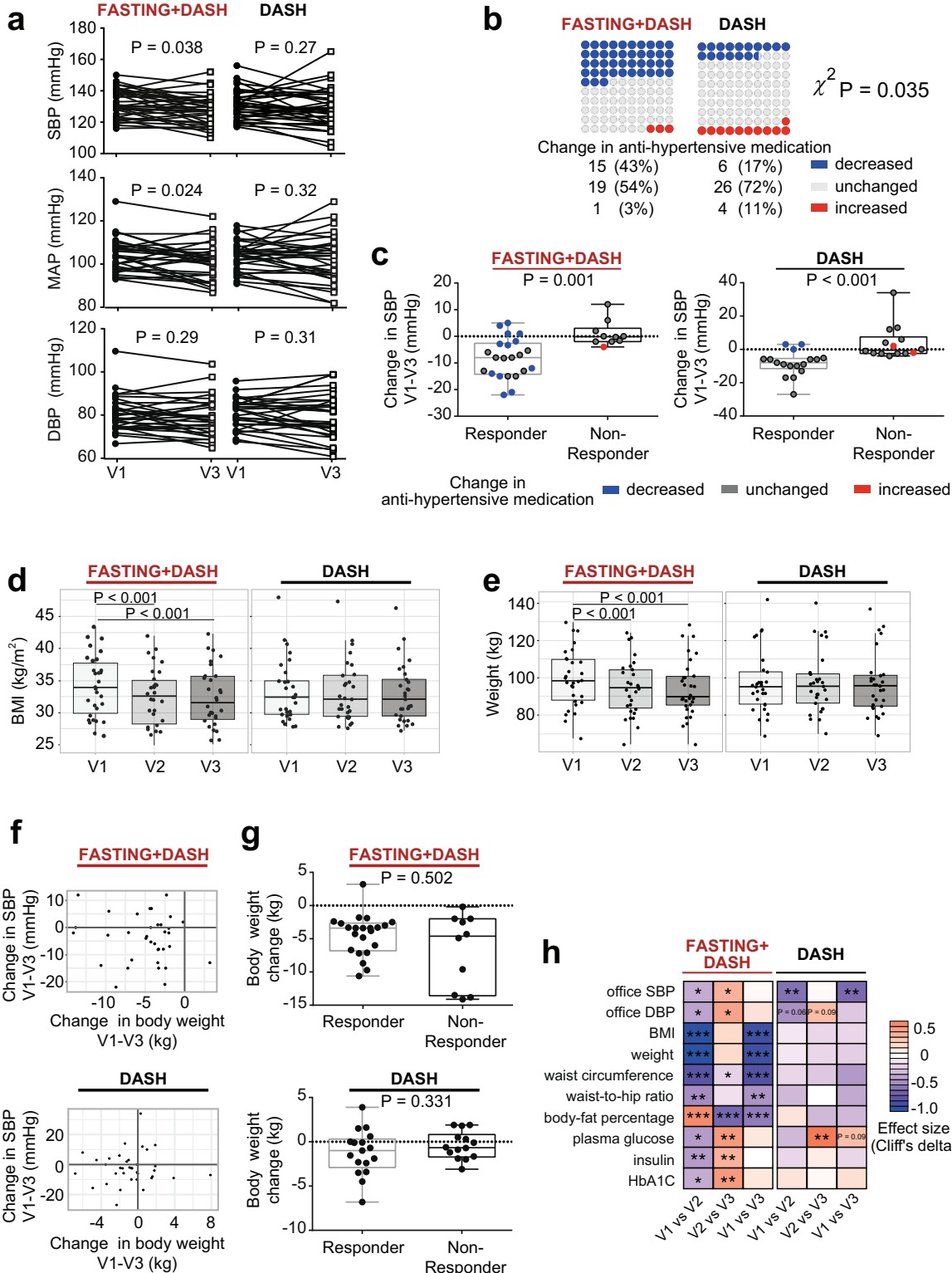

were uniquely characterized by immediate and sustained enrichment of an unclassified *Clostridium* species, with concomitant depletion of *Sphingomonas* (genus-16S) and Prevotellaceae *NK3B31* group (Fig. 4a). In addition, responders experienced a significant and sustained enrichment of the butyrate-producer *F. prausnitzii* upon refeeding (Fig. 4a). We further classified microbiomes in the fasting+DASH arm into enterotypes as previously described[15], finding a trend towards more samples shifting enterotype during intervention in subjects,

who achieved BP decrease (Supplementary Fig. 7). Virtually no overlap with effects seen in the equally powered DASH arm were found, indicating that fasting may be needed on top of a BP-reducing diet for these changes to occur (Fig. 4a, b).

In profiling the microbial metabolic potential in BP responders, we focused on gene modules curated for relevance to metabolism in the human gut (GMM)[13]. On a functional level, responder-characteristic changes resemble those in the fasting arm at large, but with even more pronounced relative enrichment for

**Fig. 2 Fasting effects are distinct from those of a modified DASH diet only, and connected to vascular health benefits. a** Fasting followed by a modified DASH diet, but not a DASH diet alone, significantly improves 24 h ambulatory SBP and MAP 3 months post-intervention (two-sided MWU, FDR-corrected P-values are shown). Lines show individual participant trajectories. **b** MetS subjects beginning a modified DASH diet post-fasting significantly reduce their intake of antihypertensive medication by 3 months post-intervention, compared to subjects beginning a DASH diet only. Two-sided $\chi^2$ test, $P = 0.035$. **c** Changes in 24 h ambulatory SBP in responders and non-responders including change in antihypertensive medication (two-sided MWU). **d**, **e** One week of fasting followed by modified DASH diet, but not DASH diet alone, caused significant (two-sided MWU, FDR-corrected P values are shown) BMI and body weight reduction in MetS patients, persisting 3 months later. **f** Comparison of changes in 24 h ambulatory SBP and body weight, respectively between baseline and follow-up in both study arms. Each dot represents an individual. **g** Body weight change is not significantly different between responders and non-responders in the fasting arm between baseline and follow-up (two-sided MWU). **h** Selected cardiometabolic risk parameters (vertical axis) altered in the fasting arm compared to the DASH arm. Heatmap hues show Cliff's delta signed effect sizes, with asterisk indicating post-hoc univariate significance after compensating for drug dosage changes (see Methods). Horizontal axis shows each time point comparison: change during fasting/week three of DASH, change during refeeding/3 months of DASH, and change during the study period as a whole. Boxplot hinges denote 25th–75th percentile. Line within the boxplot indicates median. Whiskers on (**c**, **g**) are drawn from minimum to maximum values. Whiskers on (**d**, **e**) are drawn to minimum and maximum values, but not further than 1.5 × IQR.

propionate production (MF0126, MF0121) modules (Fig. 4c). Some modules were significantly altered in abundance only in this stratified subgroup, indicating these changes strongly characterize BP responders compared to non-responders (Supplementary Data 7). For example, pyruvate:formate lyase (MF0085) is depleted during recovery only in responders.

Changes to the immunome of responders are similar to those seen in the unstratified fasting group and differ from those in the DASH arm (Fig. 4d, Supplementary Data 2, 7, 8). In the fasting arm, several immune features related to pathogen-sensing and mucosal immunity (e.g. MAIT cells, IL-17$^+$-producing Th and $\gamma\delta$T cells) changed in abundance significantly only when tested in the responder group, indicating relevant differences between responders and non-responders. Upon fasting, the frequency of both pro- and anti-inflammatory adaptive immune cells showed a stronger decrease in responders, indicating a stronger anti-inflammatory effect of fasting in responders.

**Network analysis of microbial, immune, and clinical features**. We next aimed to explain the beneficial role of fasting on BP by studying interacting microbiome-immune features through network analysis. We assessed all triplets of pairwise interactions between host clinical phenotypes, immune cell populations, and microbiome taxa or functional profiles, respectively, using modified Spearman correlations (requiring FDR < 0.1 in each comparison of two data spaces, and $P < 0.05$ in a post-hoc test accounting for the presence of the same subjects at all three time points (see Methods, Supplementary Data 9). Figure 5a shows a chord diagram constructed from these data, where the colored outer rings are lined with components from one of our three tested system spaces during fasting, refeeding, and over the full duration of the study, and the color of the connectors between factors indicate a positive or negative association (Spearman's rho). We identified a cluster of circulating cytokine-producing MAIT cells (absolute number and fraction of CD3$^+$ T cells), which positively correlated with 24 h ambulatory SBP (Figs. 5a, 6c, Supplementary Fig. 8) and MAP, but not with 24 h diastolic BP (Fig. 5a, Supplementary Data 9).

In addition, abundance of IL-2$^+$ and granulocyte-macrophage colony-stimulating factor (GM-CSF)-producing CD4$^+$ cells significantly correlated with SBP. These immune clusters showed significant interconnection to a remarkable number of microbial SCFA producers (Fig. 5a, b Supplementary Data 10), though some are rather poorly characterized. Notably, abundance of the butyrate producers *E. rectale* (ref mOTU v2 1416) and *Dorea longicatena* (ref mOTU v2 4203), and the acetate producer *Hungatella hathewayi* (ref mOTU v2 0882) negatively correlated with the abundance of the GM-CSF and IL-2-producing CD4$^+$

T cells, and with the absolute number of IFN$\gamma^+$ and TNF$\alpha$-producing MAITs, respectively.

16S analyses of the gut microbiome identified a positive correlation between the pro-inflammatory cytokine-producing MAITs and the microbial taxa Acidaminococcaceae (family), and of two *Alistipes spp.* (*shahii* and *inops*) (Fig. 5a, b). While further characterization of these taxa in the context of the gut microbiome is needed, previously published data indicated that these taxa can produce acetate, and likely butyrate and propionate as well (Supplementary Data 10).

Abundance of KEGG module M00209 (osmoprotectant transport system), reported to facilitate the uptake of nutrients mostly found in red meat[16–18], was negatively associated with IFN$\gamma^+$ and TNF$\alpha^+$ MAIT cells (Fig. 5a, b, Supplementary Data 9). Interestingly, fasting depleted various cytokine-producing MAIT cells with the most pronounced long-lasting decrease seen in IL-2$^+$TNF$\alpha^+$ producing MAIT in BP responders (Figs. 5a, b, 6a, Supplementary Data 9).

The association between MAIT cells and BMI is still a matter of debate[19]. We found in our study that the abundance of MAIT cells did not correlate with BMI, weight, waist circumference, waist-hip ratio, or body fat percentage (Supplementary Fig. 8, Data 9). Though we did find that BMI correlated with the abundance of a subset of circulating Treg-like cells (CD62L$^+$CD45RO$^-$CD25$^+$CD4$^+$), a cell type previously linked to morbid obesity in human subjects[20].

A recent publication showed non-classical monocyte enrichment in hypertensive patients[21]. Interestingly in our study, circulating non-classical monocytes were enriched upon fasting and then depleted again upon refeeding to remain below baseline levels 3 months after fasting (Figs. 1h, 5a, Supplementary Data 2). Network analysis revealed an association between non-classical monocytes, MAP and gut abundance of *Sutterella* showed an inverse correlation with non-classical monocytes (Figs. 5a, b, 6d, Supplementary Data 9).

**Baseline indicators predicting efficacy of fasting on blood pressure**. As previously stated, a large proportion of fasting patients responded with a substantial drop in BP, allowing them to reduce their use of antihypertensive medication while BP remained controlled. As not all patients experienced this beneficial effect, we sought to understand whether the factors underlying successful fasting intervention in the BP responders could be predicted at baseline. Responder and non-responder subgroups differ considerably in immunome and microbiome features, not only post-fasting and at three-month follow-up, but also at baseline, suggesting a favorable clinical response may be predictable in single patients (Supplementary Fig. 9A–C).

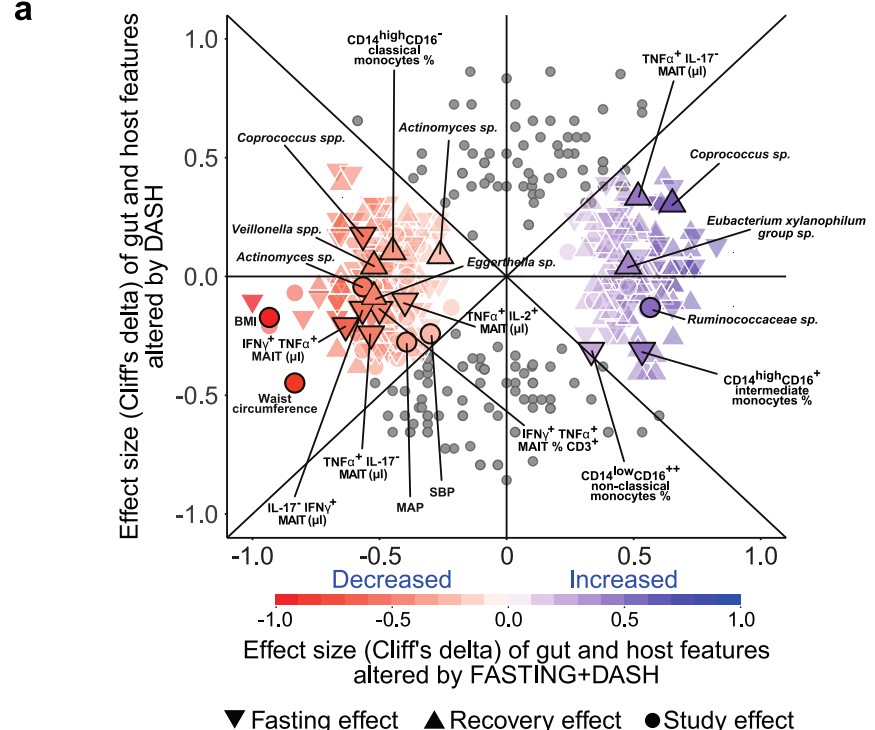

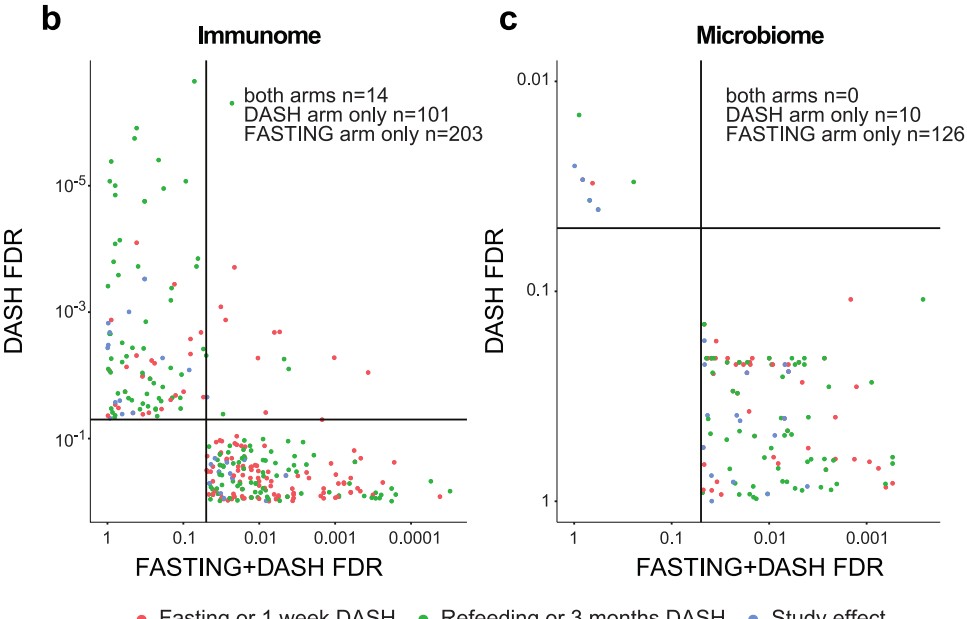

**Fig. 3 Fasting and recovery effects are not replicated in an equally powered control cohort, indicating they are intervention-specific. a** A majority of host and microbiome effects reported from the fasting+DASH arm are not replicated in DASH-only patients. Comparative effect size plot contrasting features altered significantly only under fasting+DASH (colored markers, $n = 315$) with features altered significantly also under DASH alone, or with absolute effect size greater in DASH alone (gray markers, $n = 146$). For the former category, color hue shows direction of effect, color intensity scope of effect, and marker shape which time point comparison is shown. Vertical axis shows effect size in DASH only, horizontal effect size in fasting+DASH. Selected features are named for reference. **b, c** Volcano plots show post-hoc FDR for all features significantly altered in either arm between any two time points in the fasting arm (horizontal axis), compared to the same sample number DASH arm (vertical axis). Point color shows which time point comparison is plotted. Quadrants (formed by the FDR < 0.05 thresholds) and summary counts highlight features significantly altered in each dataset for immune cell (**b**) and functional or taxonomic microbiome features (**c**). Only the fasting arm had a significant effect on the microbiome, and while a smaller fraction of immune features were altered in the DASH-only arm, these were largely not significant in the fasting arm.

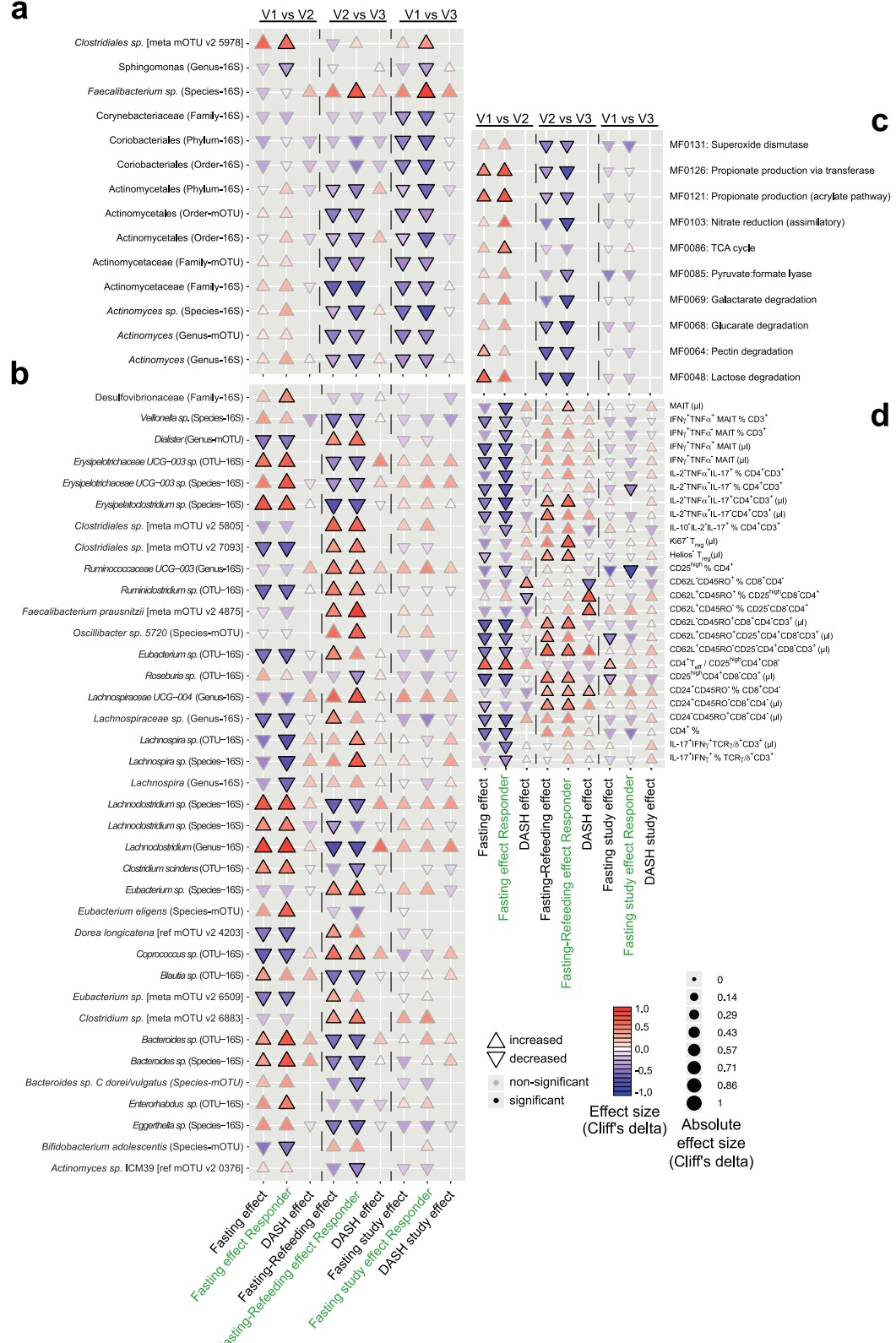

To further elucidate this phenomenon, we applied machine-learning algorithms and empirically show that we can make effective predictions from the immunome data. From 494 total immune variables, stepwise forward regression identified the top ten discriminators of responders from non-responders at baseline. Evaluating the machine-learning model, we constructed for predicting whether fasting+DASH will reduce BP by testing it on unseen data, a prediction accuracy of 71% (sensitivity 75%, specificity 70%, and F1 score 77%) was achieved using a leave-subject-out cross-validation for whether or not a future patient would respond favorably to fasting with regards to BP (Fig. 7a). Within this multivariate analysis, the driving immune features

**Fig. 4 Subjects responding favorably to fasting exhibit stronger changes in commensal abundance under intervention. a** Cuneiform plot shows subset of bacterial taxa, at different taxonomic levels, and measured either using 16S sequencing or shotgun sequencing, altered significantly (drug-adjusted post-hoc FDR < 0.05) in abundance tested in intervention responders only (vertical axis) and showing a study effect, comparing to baseline and follow-up (V3). Signed effect size are shown through marker direction and color, hue and size represent absolute effect size. Solid borders indicate significance. Markers not shown could not be tested in the DASH arm as shotgun data was unavailable, or showed no difference in rank-transformed values (Cliff's delta=0). Horizontal axis separates tests for fasting (comparison of baseline to after one week), recovery (comparison of after one week to 3 months), and study effect (comparison of baseline to 3-month follow-up). DASH results are from the DASH arm only, responders are tests using only the responders (as per decision tree) in the fasting arm. **b** Same view as (**a**), showing 16S or shotgun sequencing microbial taxa significantly altered either at fasting (V1 vs. V2) or refeeding (V2 vs. V3) in responders excluding features already in (**a**) to avoid redundancy. **c** same view as (**a**), with regards to gut functional modules (selected subset shown for clarity). **d** Same view as (**a**) but with regards to immune cell subpopulations (selected subset shown for clarity). $T_{reg}$: FoxP3$^+$ cells, MAIT: V$\alpha$7.2$^+$CD161$^+$CD4$^-$CD3$^+$.

of this classifier highlighted a lower CXCR3$^+$CD25$^-$CD4$^+$/CD25$^{high}$CD4$^+$ (most likely Th1/Treg ratio), alongside lower abundances of CD24$^+$ memory CD8$^+$ T cells and IL-17$^+$TNF$\alpha^+$MAIT cells in responders relative to non-responders (Fig. 7b, Supplementary Fig. 9E). Regarding the top ten features derived as indicative for successful patient classification, responders seem to have less of a pro-inflammatory immune signature at baseline (Fig. 7b). Notably, we could increase the prediction performance of the classifier up to 78% by using changes of immune cell abundances between baseline and 3-month follow-up visit as a basis for prediction of BP response at the single-patient level (Supplementary Fig. 9D, F). In contrast, for subjects on a DASH diet only, corresponding classifiers were unable to predict BP response above chance level.

Regarding responder-specific features, we identified microbial features as both characteristic of responders at baseline and during the intervention (Fig. 8a). Microbiomes of BP responders were depleted pre-intervention for Desulfovibrionaceae, previously shown to be enriched in type 2 diabetic patients in a Chinese cohort[17], and were moreover depleted of propionate biosynthesis genes (Fig. 8a). Fasting strongly elevated the abundance of this taxa and enriched these propionate production modules, indicating that responders suffer a treatable deficit. By 3 months post-intervention, propionate modules are almost back at baseline while BP (relative to medication dosage) remains improved, suggesting that their transient elevation during refeeding may have stabilized a less hypertensive state through mechanisms active beyond the gut (Fig. 8a). An opposing pattern was shown by a poorly characterized *Lachnospira sp.*, which had a higher abundance in responders at baseline (Fig. 8a). These findings indicate that baseline state of the gut microbiome in these MetS patients predicts individual degree of success of the fasting+DASH intervention.

The question was raised whether independent data could confirm these findings. We therefore reanalyzed the data from the only other existing cohort investigating the effect of fasting, where both BP data and stool sequencing[22] (herein referred to as "Mesnage data") was available, using the same software pipeline as for our own samples. We compared the results to ours, collapsing species/OTU fasting/refeeding/long-term follow-up signals in either dataset at the genus level for clarity (Fig. 8b, Supplementary Data 11). Despite substantial differences between the two study settings (e.g. MetS vs. healthy, mixed vs. single-sex cohort) even at reduced statistical power (Mesnage $n = 15$), we observe substantial agreement between the two datasets; dynamics of *Bifidobacterium*, *Roseburia*, *Bacteroides*, *Coprococcus* and *Intestinimonas* are comparable (Fig. 8b). Though differences can also be observed in the patterns of *Oscillibacter* and *Alistipes* in these two studies. The SCFA producer *Faecalibacterium* showed discordant fasting responses in the healthy vs. MetS cohort but exhibited consistent growth upon refeeding in both datasets (Fig. 8b).

Due to the similarity of the study designs, we next assessed whether a decrease in BP in the Mesnage cohort could be predicted by a model trained on our 16S dataset. We classified the Mesnage patients according to their BP decrease 3 months post-fasting (Supplementary Data 12). A stepwise selection model was built on our 16S baseline data, filtered for significant responder-specific taxa. The model was then evaluated, using the corresponding features from the Mesnage dataset as input. The model classified correctly 10 out of 15 subjects in the Mesnage cohort as either BP responders or non-responders. Top five contributors to the predictor highlighted gut microbiomes of non-responders to be enriched and responders to be depleted of the taxa Desulfovibrionaceae, Hydrogenoanaerobacterium, *Akkermansia*, Ruminococcaceae GCA-900066225 and *Hydrogenoanaerobacterium sp.* (Fig. 8c).

## Discussion

Here we demonstrate that fasting induces changes to the gut microbiome and immune homeostasis with a sustained beneficial effect on body weight and BP in hypertensive MetS patients. There is a growing interest in understanding how dietary interventions shape the gut microbiome and interact with metabolic diseases, including obesity, MetS, type 2 diabetes, and (cardiovascular) health[8–10,23–27]. Several lifestyle interventions aimed at weight loss have shown that the gut microbiome changes in obese, type 2 diabetic or MetS patients[10,23,24,26,27]. Although these interventions led to beneficial clinical outcomes, their effect on the gut microbiome was highly variable[10,23,24,26,27] (more information in Supplementary Data 13). In mice, intermittent fasting decreased obesity-induced cognitive impairment and insulin resistance associated with increased abundance of the *Lactobacillus* and the butyrate-producer *Odoribacter*[25]. In a small human pilot study, Ramadan fasting[9] affected the microbiome of healthy subjects enriching several SCFA producers. Each of the aforementioned studies are described in greater detail in Supplementary Data 13.

We have carried out the first high-resolution multi-omics characterization of (periodic) fasting in patients with MetS, including detailed clinical and immunophenotyping along with gut microbiome sequencing. Our major finding is that periodic fasting followed by 3 months of a modified DASH diet induces concerted and distinct microbiome and immunome changes that are specific to fasting itself, leading to a sustained BP benefit (Fig. 3a), which was not seen in the patients following a DASH diet alone.

Fasting followed by modified DASH also led to a significant long-term reduction in body weight. However, neither the change in BP nor global changes to the microbial composition or immunome appeared to be mediated by this BMI decrease (95% of findings retained significance when deconfounded for BMI change, see Supplementary Data 5, and body weight reduction was not more pervasive in treatment responders than non-

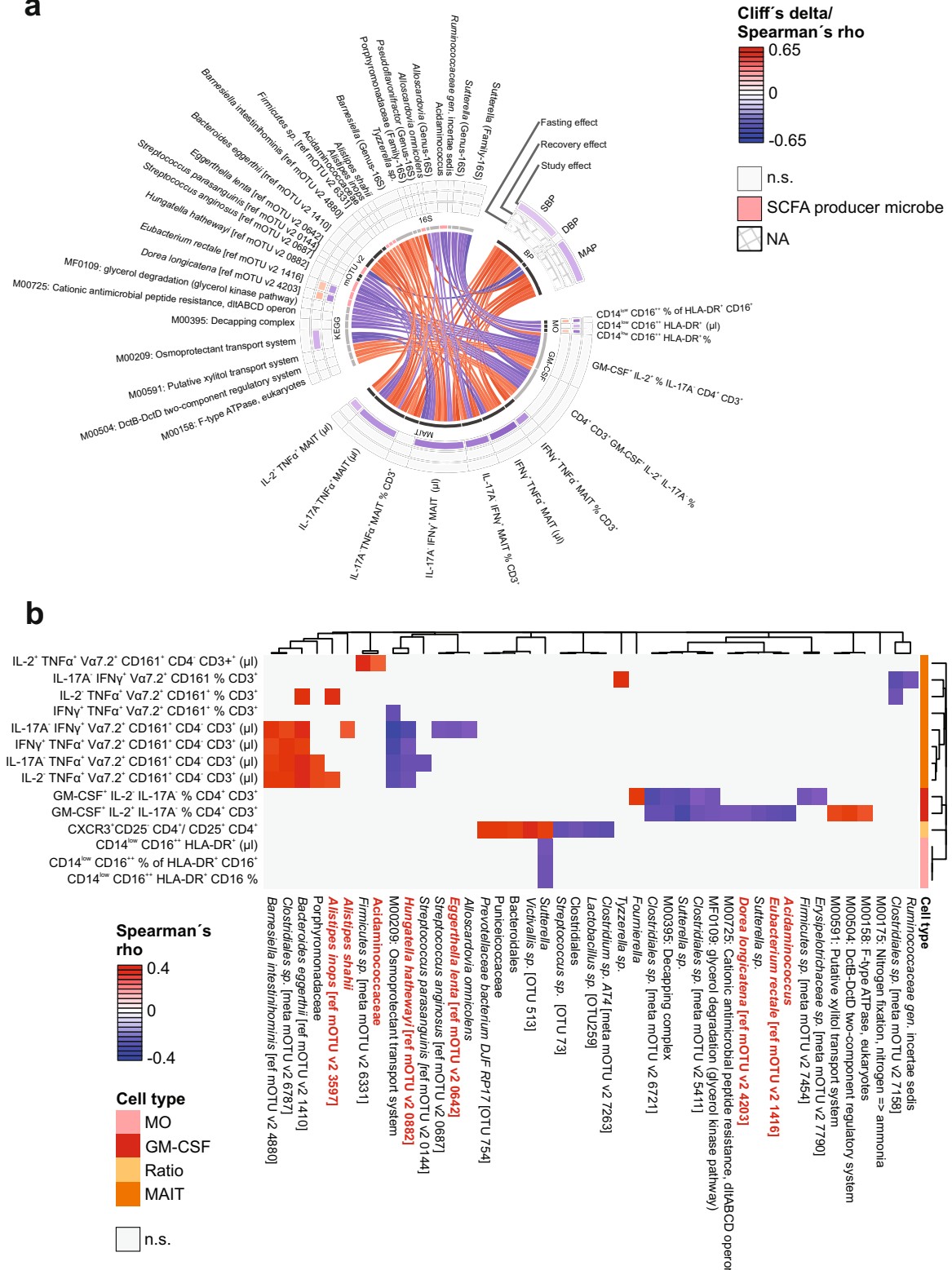

**Fig. 5 Blood pressure-microbe-immune association. a** Chord diagram visualizes the interrelation between BP (24 h ambulatory systolic, mean or diastolic BP) and fasting-impacted microbiome functional or taxonomic features, and immune cell subsets. Features are shown that form triplets of immune, microbial and phenotype variables where at least two of three correlations are significant (Spearman FDR < 0.05, post-hoc nested model test accounting for same-donor samples < 0.05) in the fasting arm of our cohort, and where in addition one or more features is significantly (drug-adjusted post-hoc FDR < 0.05) affected by the intervention. Color of the connectors indicates positive or negative association (Spearman's rho), color of the cells within the tracks indicates changes upon fasting, refeeding and study effect (Cliff's delta, white if not significant), respectively. **b** Hierarchical clustering of microbiome features-associated immune features. Color indicates Spearman's rho.

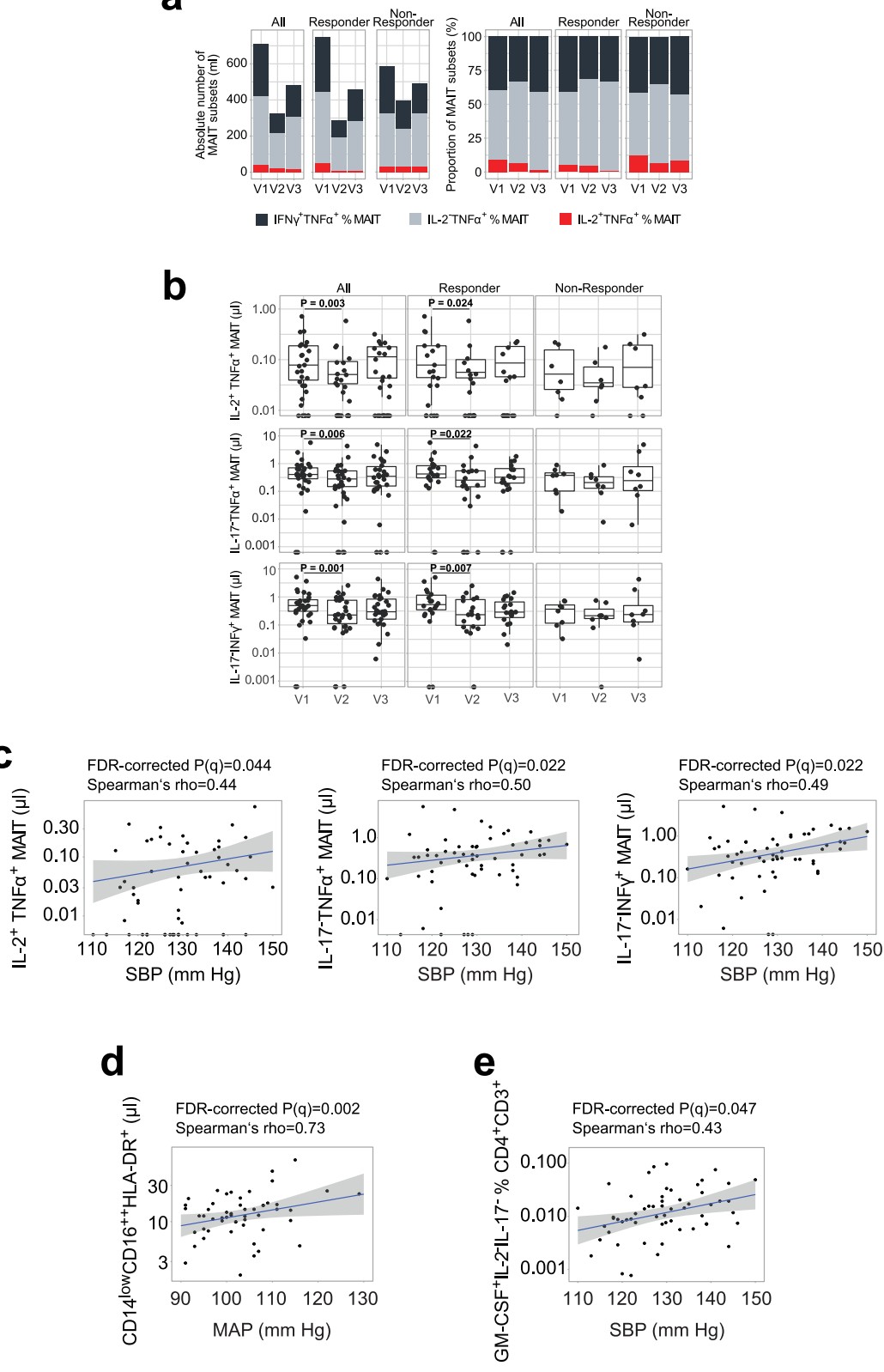

responders, Fig. 2g). Furthermore, BP and BMI were both associated with various immune cell subsets and microbial taxa on a multivariate level, and the effects of fasting on these two features are divergent (shown as chord plots on Fig. 5, Supplementary Fig. 8, respectively). Nevertheless, the data indicate that a 5-day fast exerted an effect on microbiome composition and immune cell subsets. Even though many of these shifts post-fasting are

transient, a sustained improvement of BP was seen in our patients. Comparison of V1 to V2 suggests that microbiome and immune cells may reset to some extent during and after the intense caloric restriction, similar to a preconditioning mechanism. The subsequent DASH diet consistent across all patients thus seem to act differently depending on whether this preconditioning took place or not. This interpretation is supported by the fact

**Fig. 6 The association between blood pressure and specific circulating immune cell populations. a** Cumulative absolute number and relative abundance of circulating IFNγ⁺TNFα⁺, IL-2⁻TNFα⁺ and IL-2⁺TNFα⁺ mucosa-associated invariant T cells (MAIT) cells within the fasting arm subdivided by BP responsiveness (median, n = 30 for all, n = 20 for responders, n = 8 for non-responders, respectively). Absolute number of circulating IL-2⁺TNFα⁺ (All: $P = 0.019$, Responder: $P = 0.024$), TNFα⁺ (All: $P = 0.006$; Responder: $P = 0.022$) and IFNγ⁺ (All: $P = 0.001$; Responder: $P = 0.007$); (**a**) two-sided MWU test after Benjamini–Hochberg correction. **b** MAIT cells within the fasting arm subdivided by BP responsiveness (n-number as in (**a**); two-sided MWU test after Benjamini–Hochberg correction. **c** Correlations of circulating IL-2⁺TNFα⁺, TNFα⁺, and IFNγ⁺ MAIT cells and 24 h ambulatory SBP (*FDR-corrected $P = 0.044$, 0.022, and 0.022, respectively). **d** Correlations of circulating non-classical CD14^lowCD16⁺⁺HLA-DR⁺ monocytes and 24 h ambulatory MAP in responder (FDR-corrected $P = 0.002$). **e** Correlations of circulating GM-CSF⁺IL-2⁻IL-17⁻ of % CD3⁺ and 24 h ambulatory SBP (FDR-corrected $P = 0.047$). n-number for (**c–g**) as in (**b**). MAIT: Vα7.2⁺CD161⁺CD4⁻CD3⁺. Boxplot hinges denote 25th–75th percentile. Line within the boxplot indicates median. Whiskers are drawn to minimum and maximum values, but not further than 1.5 × IQR. **c–e** Gray shading represents 95% CI.

that the DASH diet alone neither reduced SBP nor BMI, while affecting different (and substantially fewer) immune cell subsets. In line with the preconditioning hypothesis, we consider that (1) those subjects who benefit most with regards to BP from a fasting+DASH intervention are those depleted at baseline for both SCFA producing taxa (including core butyrate producers) and SCFA production gene modules; (2) that such taxa and gene modules enrich either during the fasting phase or the refeeding phase thus ameliorating the aforementioned baseline depletion; and (3) that at least some enrichment remains at 3-month follow-up in BP responders (less so in non-responders). Our interpretation is that one crucial mechanism for the improvement stems from the effects of increased SCFA availability, either locally in the intestine (impacting immune signaling and intestinal permeability), systemically, or both. While we cannot directly test it in the present cohort, it is a scenario consistent both with expectations from the literature and with our observations of a consistent depletion-then-regrowth pattern. Thus, future work will include studying a larger fasting/refeeding cohort at various intermediate time intervals.

Fasting induced a profound change in circulating immune populations; e.g. depleted Th1 cells and permanently enriched dendritic cells, which both have been shown previously to play a role in the pathogenesis of experimental hypertension[28,29]. Further, we discovered significant correlations between circulating MAIT cells and 24 h ambulatory BP and MAP.

A growing body of evidence suggests that the abundance of certain microbes is associated with cardiovascular health. Previous reports on hypertensive patients have shown taxonomic and functional gut microbiome shifts[6,7]. For example, Firmicutes have been shown to be more abundant in healthy controls compared to pre-hypertensive and hypertensive patients[7]. Upon fasting, several Clostridial Firmicutes shifted significantly in abundance, with an initial decrease in butyrate producers such as *F. prausnitzii*, *E. rectale* and *C. comes*, which were reverted after 3 months upon refeeding; with the latter taxon likely being an indirect effect of the observed weight reduction (Supplementary Data 5). Further, functional microbial metabolism in fasting patients at baseline share some similarities to the previously profiled hypertensive microbiome[7]. In the fasting arm, the functional shift during refeeding enriches for functional modules also enriched in non-hypertensive controls, i.e. for potentially BP-protective factors.

Clinical studies represent a highly heterogeneous situation with multifactorial disease features and strongly variable microbial and lived environments. To account for this heterogeneity, we compared the data from our longitudinal study (post-fasting and 3-month) to the respective baseline values of the study subjects. This intraindividual analysis allowed us to identify BP responder-specific changes in spite of the reduced power in such a substratified analysis. The responder-specific microbiome changes in our fasting arm post-intervention (enrichment of *F. prausnitzii*, *Bacteroides* and Firmicutes, depletion of *Actinomyces*) are likely

beneficial to the host. A recent study profiling the hypertensive microbiome showed that during disease, patients experienced an enrichment of Actinomyces, and a depletion of *F. prausnitzii*, *Bacteroides* and Firmicutes[7]. Moreover, Guevara-Cruz et al. recently showed in a Mexican cohort involving 146 MetS patients, that a 75 day long, 500 kcal/day, low-saturated fat dietary intervention improved the clinical phenotype, significantly decreased gut dysbiosis and increased the abundance of *Akkermansia muciniphila* and SCFA producer *F. parusnitzii*[27] (Supplementary Data 13). Furthermore, abundance of some functional gut-specific gene modules was significantly altered in our dataset only in BP responders, for example, the pyruvate:formate lyase module, MF0085, which was decreased after refeeding. This decrease (from a trending elevation at baseline) may contribute to vascular health, as a recent study demonstrated enrichment of the same enzyme in atherosclerosis patients relative to healthy controls[30], and formate production has been previously linked to BP regulation[31,32].

Stratification of the cohort to BP responsiveness showed that also immune changes present in the fasting arm are more pronounced in responders than in non-responders, and are fundamentally different from the changes observed in the DASH-only arm. The DASH-only arm was associated with the decrease of CD8⁺ Tem cells, previously reported to play a role in hypertension[29,33]. Responders and non-responders not only reacted differentially to fasting, but also differed at baseline with regards to their propionate synthesis capacity pre-intervention and the relative depletion by depletion of Desulfovibrionaceae, which has been linked to a lean phenotype[34,35]. These features were then normalized during fasting. Notably, recent experimental work suggested an antihypertensive effect of propionate treatment in mice[36]. Furthermore, responders were enriched in *Lachnospira sp.* at baseline, which was shown to contribute to diabetes in obese mice and is enriched in obese children[37,38]. Our findings indicate responders and non-responders to our intervention differ with regards to several gut microbiome features relevant to hypertension, with fasting-induced normalization of these differences seen during a successful fasting intervention.

Through network analysis of the immunome, microbiome, and clinical data, we identified significant correlations between circulating MAIT cells and 24 h ambulatory SBP and MAP. MAIT cells represent up to 10% of peripheral blood T cells, but in contrast to other classical T cells[29], have not yet been linked to the regulation of BP. They differ in many aspects from conventional T cells by expressing a semi-invariant TCR α-chain Vα7.2-Jα33. MAITs can produce various cytokines mimicking an effector/memory-like phenotype and yet they behave rather like innate cells. During aging[18] and CMD[19,39], absolute circulating MAIT number and frequencies decrease, while certain subsets of cytokine-producing and adipose tissue MAITs were found to be enriched in obese type 2 diabetic patients[19]. In addition, this network analysis revealed that abundance of SCFA producing microbes correlates significantly with circulating

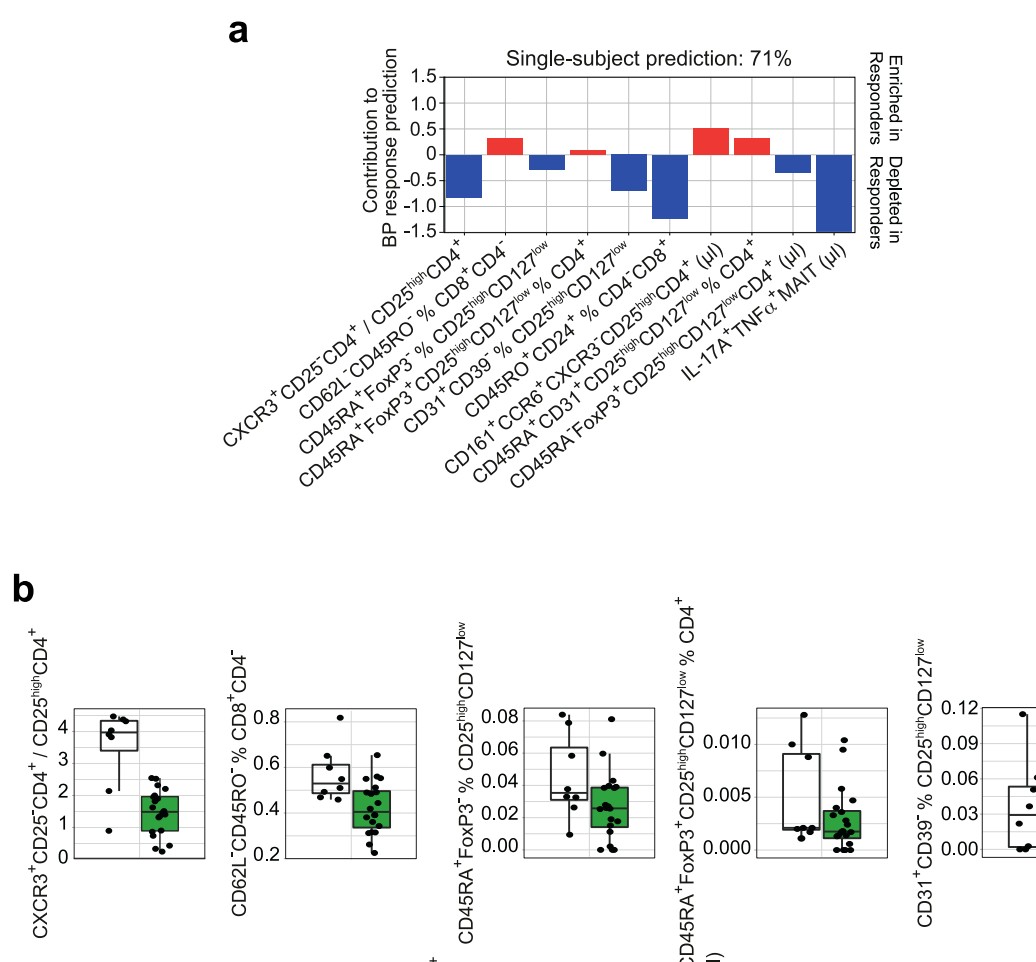

**Fig. 7 Long-lasting BP responders and non-responders differ in immunome composition. a** Prediction model weights for BP response using the immunome dataset at baseline. The top ten immunome features were used to build a multivariate logistic-regression algorithm. Single-subject prediction was quantified using a leave-one-out cross-validation procedure. The bar plots represent the regression in a model with binary output (responder yes = 1 vs. no = 0) for every feature. **b** Quantification of the immunome features at baseline used in the prediction model to predict BP response in the future, split into responders and non-responders. MAIT: $V\alpha7.2^+CD161^+CD4^-CD3^+$. Boxplot hinges denote 25th–75th percentile. Line within the boxplot indicates median. Whiskers are drawn to minimum and maximum values, but not further than $1.5 \times IQR$.

pro-inflammatory cytokine-producing MAIT cells and GM-CSF$^+$ IL-2$^+$ T helper cells. Of note, most of these microbes are relatively poorly characterized taxa and further description is needed to elucidate their role in the gut and as contributors of dys- or eubiosis.

Using machine learning, we were able to utilize deep immunophenotyping data to predict at baseline, which subjects were likely to decrease their BP during fasting despite the small number of subjects. In addition, the accuracy of the prediction was enhanced further taking the dynamics of immune populations along the course of the study into account. No

corresponding prediction of a favorable response to a DASH-only intervention was possible. The features informing the predictor indicate BP responders and non-responders present with differing severities of a pro-inflammatory immune signature at baseline, raising the question whether responders and non-responders suffer from varying degrees of MetS severity at baseline. Remarkably, no significant difference in baseline BP, BMI, lipid levels, or glucose homeostasis parameters between BP responders and non-responders was observed before the intervention (Supplementary Data 6). However, BP responders exhibited higher median SBP than non-responders (135 mmHg and 128 mmHg,

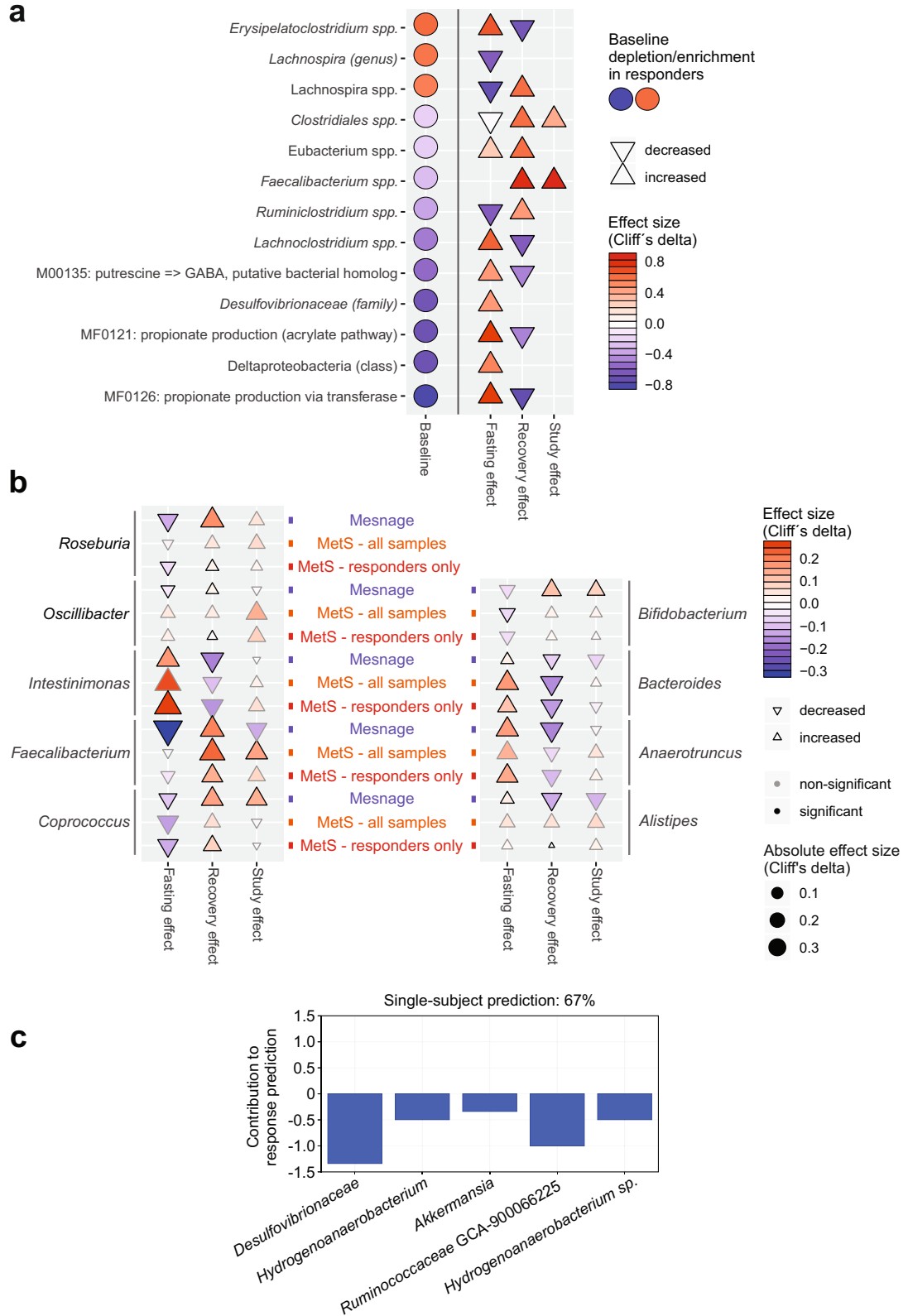

respectively). Baseline antihypertensive medication did not differ significantly between the groups (responders' normalized mean dose: 1.4, non-responders' normalized mean dose: 2.1). Additionally, responders had lower median BMI than non-responders; 32.0 and 36.5, respectively. In addition, body fat percentage was slightly higher in the fasting+DASH group compared to the DASH group (median 42%, 39%, respectively). Furthermore, BP responders had a baseline median LDL of 149 mg/dl compared to 122 mg/dl for non-responders, while HDL did not differ (in both groups 48 mg/dl). These data indicate that although BP

**Fig. 8 Baseline microbiome predicts long-lasting BP responsiveness. a** Circles denote features differing at baseline in responders vs. non-responders and altered during intervention in responders. Effect size (Cliff's delta) is shown comparing responders and non-responders. **b** Comparison of results from the present study (MetS; all samples and BP responders only shown as orange and red tags, respectively, separately) with those of a recent similar fasting intervention in healthy men (Mesnage; blue tags). Effect sizes at the species or OTU level were averaged at the genus level for clarity, and are shown in the plot (direction rendered as marker shape and hue; scope rendered as marker size and intensity) for all genera where at least one constituent taxon achieved significance either in the Mesnage or MetS study (these are shown in boldface). Columns denote phases of each intervention - fasting phase, refeeding, and follow-up vs. baseline. Substantial agreement between the two studies is seen, which is typically stronger for the subset of BP responders. **c** Prediction model weights for BP response using the MetS 16S dataset at baseline. The top five immunome features were used to build a multivariate logistic-regression algorithm. Single-subject prediction on the Mesnage dataset[22] was quantified using a leave-one-out cross-validation procedure. The bar plots represent the regression in a model with binary output (responder yes = 1 vs. no = 0) for every feature.

responders and non-responders do demonstrate slightly different trends in some clinical parameters, BP responders do not show any less severe disease phenotype.

Through the reanalysis of the Mesnage dataset, the only fasting cohort in the literature with a similar study design and which includes both BP and microbiome data, we were able to demonstrate concordant treatment-related microbiome shifts in both studies. This finding suggests the effects of fasting and refeeding on gut microbiota generalizable. A machine-learning model built from microbiome features differentially abundant at baseline in BP responders in our cohort was able to predict significant long-term BP decrease in the Mesnage et al. subjects with about 70% accuracy, further supporting the idea that these findings are likely generalizable.

Previous works have also shown that some outcomes of dietary interventions in cardiovascular patients might be related to baseline microbiome features. Notably, a recent study demonstrated that higher baseline *Akkermansia* abundance was associated with persistent weight loss in a study investigating MetS/obese patients undergoing a 52-week long weight reduction program[10] (Supplementary Data 13). In addition, Velikonja et al. showed in a study investigating the effect of beta-glucan supplementation in MetS patients that a higher baseline abundance of *Akkermansia muciniphila* and *Bifidobacter* spp. was characteristic of patients whose cholesterol decreased due to the intervention[23] (Supplementary Data 13).

Thus, we demonstrate the practical utility of a machine-learning analysis pipeline for predicting BP benefit of fasting in MetS patients with hypertension using both baseline immunome and microbiome data.

It is important to recognize that our study represents patients with hypertension and MetS solely from a Caucasian-European background. This selection criterion introduces a selection bias in our study design. Additional research is necessary to elucidate whether the results presented here could be applicable in a more heterogeneous patient population. Further, our recruitment procedure could already have introduced a selection bias toward patients who were interested in fasting/dietary studies and therefore are sensitive about their cardiometabolic health. Since the participants were especially interested in the fasting procedure, the allocated DASH participants were offered a cost-free fasting cycle after successful completion of the study. However, we cannot exclude that this led to an increased long-term motivation compared to the participants who started with the fasting protocol. Furthermore, the study design did not allow us to investigate the long-term effects of a fasting intervention without a subsequent DASH diet on the BP, microbiome, or immunome. In our cohort, fasting was required on top of DASH to achieve the observed outcomes, but we cannot conclude (and do not expect) fasting without a subsequent dietary change to do so either. We can only claim fasting was required prior to the DASH diet to achieve the effects observed in our cohort. DASH, which is rich in fibers, might furthermore "fuel" the beneficial microbiome, thus

further contributing to cardiovascular health, and may play a part in maintaining this microbiotal state longer. However, some effects are replicated in the similar dataset from healthy males (without MetS and without DASH intervention) in the Mesnage dataset[22], thus indicating the precise DASH setup may not be strictly needed. Most likely, the two components of the intervention synergize—fasting may potentiate the microbiome in these patients to be shifted to a more DASH-compatible microbiota upon diet change. While we identify changes in microbial taxonomic and functional features, bacterial metabolites and immune processes, which could explain the efficacy of the intervention, robust conclusions of causality will require follow-up experimental work, particularly in animal models (e.g. gnotobiotic mice colonized with bacteria strongly associated with BP). In addition, the relatively low patient number could be regarded as a limitation. Although our present study is large enough to allow inference of significance for the strongest contributors to the observed effect, our results are likely not complete, and follow-up in additional and larger studies will be needed for a comprehensive view of subtle fasting-associated host and microbiome features. Our study design did not allow for the blinding of participants regarding their intervention. To maximally reduce the bias, the scientific staff were blinded during the course of processing, measurement, and analysis of collected samples. Further, the present study cannot infer how frequently fasting cycles should be repeated to control BP in at-risk patients, nor whether it is as effective without a concomitant DASH intervention. Despite the low number of participants of the study, machine-learning algorithms were able to predict BP responsiveness based on the immunome and 16S data. Only the latter could be confirmed in an independent dataset, as no equivalent immunome profiling in a fasting dataset has been published to date. Confirmation of the predictive capability of the immunome data and testing further hypothesis raised above (e.g. the interaction between SCFA availability and BP responsiveness) require future prospective clinical studies. The favorable impact of fasting followed by a DASH diet during refeeding phase shown here highlights this intervention as a promising non-pharmacological intervention for the treatment of high BP in MetS patients.

## Methods

**Study planning and ethical approval.** The study was planned as part of a randomized-controlled bi-centric trial conducted by the outpatient center of the department of Internal and Integrative Medicine at Charité-Universitätsmedizin. The study was approved by the ethics committees of the Charité-Universitätsmedizin Berlin (approval number: EA4/141/13) and registered at ClinicalTrials.gov (registration number: NCT02099968).

**Participants.** Participants were recruited from the existing patients at study centers and through local newspaper announcements. Patients were first screened over the phone by a research assistant to assess eligibility. Eligible patients were invited for an assessment by a physician, where they were examined and provided detailed written information describing the study. If patients met all inclusion criteria and did not meet any exclusion criteria, informed consent was obtained and they were included in the study.

Male and female patients with MetS according to National Cholesterol Education Program Adult Treatment Panel III (NCEP ATP III) criteria were included. MetS was defined as the presence of at least three out of five risk factors: (i) increased waist circumference (>94 cm in men and >80 cm in women), (ii) hypertriglyceridemia (>150 mg/dl (1.7 mmol/l) or lipid-lowering medication), (iii) low levels of high-density lipoprotein cholesterol (HDL-C; < 40 mg/dl (1 mmol/l) in men and <50 mg/dl (1.3 mmol/l) in women or use of HDL-increasing medication (niacin or fibrate), (iv) elevated blood pressure (≥130/85 mm Hg or use of antihypertensive medication), and (v) elevated fasting plasma glucose (≥110 mg/dl or treatment for diabetes mellitus). Beyond NCEP ATP III criteria, patients were required to have been diagnosed with systolic hypertension (either being on antihypertensive medication or untreated). Further inclusion criteria included basic mobility and the ability to provide informed consent.

Exclusion criteria included (i) diabetes mellitus type 1 or insulin bolus therapy (c-peptide < 1.2 ng/ml), (ii) manifest treated coronary artery disease, myocardial infarction, pulmonary embolism, or stroke within the past 3 months, (iii) heart failure ≥ stage I NYHA, (iv) peripheral artery disease ≥ stage 2, (v) chronic kidney disease > stage 2 (GFR < 60 ml/min), (vi) manifest eating disorder, vii) dementia or manifest psychosis, or viii) other severe internal diseases.

## Periodic fasting and plant-based Mediterranean diet intervention

*Dietary interventions.* The interventions in both groups were delivered as an intensive group-based behavioral intervention. The educational concept incorporated aspects of the mind–body program designed by the Benson–Henry Mind/Body Medical Institute of Harvard Medical School[40]. The dietary education included counseling, comprehensive lectures and cooking classes.

*Periodic fasting and modified DASH diet intervention.* Intervention within the fasting arm (Fig. 1a) started with two calorie-restricted vegan days (max 1200 kcal/day), followed by 5-days with a daily nutritional energy intake of 300–350 kcal/day, derived from vegetable juices and vegetable broth. After completion of fasting, weekly 6 h multimodal sessions were provided for a total of 10 weeks; both groups received intensified nutritional counseling/nutritional classes and additional general lifestyle recommendations for exercise and stress reduction[41]. The program entailed 10 h of group sessions for the initial periodic fasting and 50 h of nutritional education, which included lectures and cooking lessons. Similar to protocols from previous trials on periodic fasting in rheumatoid arthritis and diabetes mellitus type 2[42,43] patients were instructed to follow a modified DASH diet after the fasting period, with additional emphasis on plant-based and Mediterranean diet to optimize refeeding[44–46].

*Modified DASH diet intervention.* The DASH group (Fig. 1a) was trained in the Dietary Approaches to Stop Hypertension (DASH) diet, a sodium-, fat- and sugar-reduced mainly plant-based diet, which has been shown to reduce high blood pressure[47,48]. The intervention was similarly delivered as an the fasting group-based behavioral intervention with aspects of the mind–body program of the Benson–Henry Mind/Body Medical Institute, Harvard Medical School[40]. Overall, the program consisted of 50 h of group sessions over a period of 10 weeks and also included comprehensive lectures and cooking lessons.

*Randomization.* Patients were randomly allocated to Fasting or DASH by block-randomization with randomly varying block lengths, stratified by a) study center, and b) the intake/non-intake of antihypertensive medication. The randomization list was created by a biometrician not involved in patient recruitment or assessment using the Random Allocation Software[49]. The list was password-secured and only the biometrician was able to access it. On this basis, sealed, sequentially numbered opaque envelopes containing the treatment assignments were prepared.

*Outcome measures.* Outcomes were assessed at baseline and at 1 and 12 weeks after randomization by a blinded outcome assessor who was not involved in patient recruitment, allocation, or treatment. Two primary outcome measures were defined: 24 h ambulatory systolic blood pressure at week 12 and the Homeostasis Model Assessment (HOMA)-index at week 12.

*Physician-assessed outcomes.* Twenty-four-hour ambulatory blood pressure monitoring (ABPM) and pulse pressure recording were performed using a digital blood pressure monitor (Mobil-O-Graph® PWA, I.E.M., Stolberg, Germany)[50]. Baseline ABPM measurements were performed within one week before the starting of the intervention, those at week 12 within a week after the end of the intervention. ABPM was initiated at the same time of day for each successive visit. The monitoring software automatically removed incorrect measurements using built-in algorithms. Blood pressure and heart rate values were further categorized as day or night values using each patient's reported awake and sleep times. Office blood pressure was measured in the hospital by a sphygmomanometer, using the average of three consecutive measurements after 5 min rest while sitting in a quiet room. Office blood pressure was measured at each time point, ambulatory blood pressure only at baseline and week 12.

Body weight, body fat percentage, and lean mass percentage were measured using the Omron BF 511 bioelectrical impedance device[51]. BMI was calculated as the weight in kilograms divided by the square of height in meters. Waist

circumference was measured by two research assistants using a measuring tape in the horizontal plane exactly midway between the iliac crest and the costal arch. Measures were repeated twice and the mean of both measures was used. If the two measures differed by more than 1 cm, both measures were repeated. Hip circumference was measured in the horizontal plain at the maximal circumference of the hips or buttock region above the gluteal fold, whichever is larger, using the same approach as for waist circumference. Waist-hip-ratio was measured as the quotient of waist circumference and hip circumference[52].

*Laboratory measures.* Blood samples were collected from the antecubital vein into vacutainer tubes and analyzed using the Modular P analyzer (Roche, Mannheim, Germany). Metabolic parameters included plasma and blood glucose levels, blood insulin levels, HbA1C, and HbA1C IFCC and were analyzed using standard procedures. HOMA index was calculated as blood insulin level (μU/ml) × blood glucose level (mmol/l)/22.5[53]. Further laboratory parameters included blood lipid levels (total cholesterol, HDL cholesterol, LDL cholesterol, LDL/HDL ratio, triglyceride), uric acid, blood creatinine level, estimated glomerular filtration rate (eGFR), C-reactive protein (CRP), insulin-like growth factor 1 (IGF-1), and interleukin-6 (IL-6), triglyceride, fasting glucose level[54]. Samples were destroyed after the analysis and were not further stored.

*Safety.* All adverse events occurring during the study period were recorded. Patients experiencing adverse events were asked to see the study physician to assess their status and initiate any necessary response. The most common symptoms during the fasting period were mild weakness, headaches, and mild perception of hunger. No serious adverse effects were reported. During the normocaloric diet periods no adverse effects were reported.

*Multiple imputation.* All analyses were conducted on an intention-to-treat basis, including all participants being randomized, regardless of whether or not they gave a full set of data or adhered to the study protocol. Missing data were multiply imputed by Markov chain Monte Carlo methods[55,56].

*Peripheral blood mononuclear cell analysis.* Whole blood staining was performed using antibodies against major leukocyte lineages. Quantitative measurement was performed using a high throughput sampler (BD) and a BD FACS CantoII (BD). Peripheral venous blood was obtained and mononuclear cells were isolated within 24 h of collection by density gradient centrifugation using Biocoll and cryopreserved until further processing. Thawed cell aliquots were either labeled for extracellular antigens using fluorophore-conjugated monoclonal antibodies or CD4⁺ cells were selected (Miltenyi CD4⁺ Selection Kit). Cells ($10^6$) from CD4⁺ and CD4⁻ fractions were placed onto U-bottom plates and re-stimulated for 4 h at 37°C and 5% $CO_2$ in a humidified incubator in a final volume of 200 μl RPMI 1640 (Sigma) supplemented with 10% FBS (Merck), 100U/ml penicillin (Sigma), 100 mg/ml streptomycin (Sigma), 50 ng/ml phorbol 12-myristate 13-acetate (PMA, Sigma), 250 ng/ml ionomycin (Sigma) and 1.3 μl/ml Golgistop (BD). After re-stimulation, cells were labeled with Life/Dead Fixable Aqua Dead Cell Stain Kit, for 405 nm excitation (Invitrogen), followed by labeling with surface antigen-specific fluorophore-conjugated monoclonal antibodies. Cells were then fixated and permeabilized by FoxP3/Transcription Factor Staining Kit (eBioscience), and subsequently labeled with intracellular-antigen-specific fluorophore-conjugated monoclonal antibodies. Antibodies are listed in Table 2. Samples were analyzed using the FACSCanto II multicolor flow cytometer (BD). The acquisition was performed with Diva 6.1.3 (BD). Data analysis was performed using FlowJo 10.3 (FlowJo LLC) and FCSExpress V6.02 (De Novo Software) software. Absolute cell numbers were calculated using the relative percentage of cell population compared to a marker used in the whole blood staining.

*FlowSOM.* Data were manually gated on single live cells and exported as FCS files in FCS Express V6.02 (De Novo Software). The automated analysis of FCS files was done by the FlowSOM[57] algorithm, an R[58] bio-conductor package that uses self-organizing maps for dimensional reduction and visualization of flow cytometry data. All data were scaled and log-transformed on import. Cells were assigned to a Self-Organizing Map (SOM) with a 10 × 10 grid, grouping similar cells into 100 nodes. Each node in the FlowSOM tree gets a score indicating its correspondence with this requested cell profile. To visualize similar nodes in branches, a minimal spanning tree (MST) was constructed and cell counts were log scaled. To visualize the differences between the two-time points, the mean percentage per sample group was computed in each cluster and then the statistical difference was performed by applying MWU test on every node within metaclusters. P values were two-sided and analysis was performed using RStudio (version 3.4.4). The FlowSOM algorithm was run 3 times to ensure reproducibility of the results and $P < 0.05$ was considered to be statistically significant.

*Medication data collection and cleanup.* Antihypertensive drugs were normalized in order to track changes during intervention. In a first step, antihypertensives (according to the WHO ATC classification system), diuretics, beta-blocking agents, calcium channel blockers, and agents acting on the renin-angiotensin system as well as the given dosage were identified at V1 and at follow-up visit after 3 months (V3).

**Table 2 Antibodies used for the flow cytometry analysis.**

| Antibody | SOURCE | RRID | Dilution |
|---|---|---|---|
| a-CD11c APC | Miltenyi | AB_871587 | 2:25 |
| a-CD123 PE | Miltenyi | AB 244211 | 1:10 |
| a-CD127 PE-Vio770 | Miltenyi | AB_2659856 | 1:10 |
| a-CD14 APC | Miltenyi | AB_244301 | 1:25 |
| a-CD14 PE-Vio770 | Miltenyi | AB_2660180 | 1:25 |
| a-CD16 FITC | Miltenyi | AB_2655402 | 1:10 |
| a-CD16 PE | Miltenyi | AB_2655404 | 1:10 |
| a-CD161 FITC | Miltenyi | AB_871631 | 1:10 |
| a-CD19 PE | Miltenyi | AB_244223 | 2:25 |
| a-CD196 (CCR6) APC | Miltenyi | AB_2655933 | 1:10 |
| a-CD24 PerCP-Vio700 | Miltenyi | AB_2660665 | 1:10 |
| a-CD25 APC | Miltenyi | AB_871644 | 1:10 |
| a-CD25 PE | Miltenyi | AB_244320 | 1:10 |
| a-CD27 PerCP-Vio700 | Miltenyi | AB_2660841 | 1:10 |
| a-CD27 PE-Vio770 | Miltenyi | AB_2660837 | 1:10 |
| a-CD3 PerCP-Vio700 | Miltenyi | AB_2659948 | 1:10 |
| a-CD31 FITC | Miltenyi | AB_871662 | 1:10 |
| a-CD39 APC-Vio770 | Miltenyi | AB_2660873 | 1:10 |
| a-CD4 FITC | Miltenyi | AB_871682 | 1:10 |
| a-CD4 VB | Miltenyi | AB_10829954 | 1:10 |
| a-CD45 FITC | Miltenyi | AB_244234 | 1:10 |
| a-CD45RA PerCP-Vio700 | Miltenyi | AB_2660987 | 1:10 |
| a-CD45RO FITC | Miltenyi | AB_10827692 | 1:10 |
| a-CD56 APC | Miltenyi | AB_244331 | 1:10 |
| a-CD62L APC | Miltenyi | AB_244246 | 1:10 |
| a-CD69 APC | Miltenyi | AB_615096 | 1:10 |
| a-CD8 FITC | Miltenyi | AB_244336 | 1:10 |
| a-CD8 PE-Vio770 | Miltenyi | AB_10829189 | 1:10 |
| a-CXCR3 PE-Vio770 | Miltenyi | AB_2655740u | 1:10 |
| a-FoxP3 PE | Biolegend | AB_10579944 | 1:5 |
| a-GM-CSF PE | Miltenyi | AB_2572656 | 1:20 |
| a-Helios FITC | eBioscience | AB_2572656 | 1:120 |
| a-HLA-DR PerCP-Vio700 | Miltenyi | AB_10839556 | 1:10 |
| a-IFNγ PE-Vio770 | Miltenyi | AB_2661063 | 1:30 |
| a-IL10 PE-Cy7 | eBioscience | AB_2573523 | 1:20 |
| a-IL17A APC-Vio770 | Miltenyi | AB_2659812 | 1:10 |
| a-IL2 FITC | eBioscience | AB_2572512 | 1:20 |
| a-IL2 PE | Miltenyi | AB_244197 | 1:10 |
| a-IL22 eFluor450 | eBioscience | AB_11150956 | 1:20 |
| a-IL5 APC | Biolegend | AB_315330 | 1:20 |
| a-Ki67 APC | Miltenyi | AB_2573218 | 1:120 |
| a-TCRγδ APC-Vio770 | Miltenyi | AB_2654040 | 1:10 |
| a-TCRγδ PE | Miltenyi | AB_2654034 | 1:10 |
| a-TCRVα 7.2 APC-Vio770 | Miltenyi | AB_2653673 | 1:10 |
| a-TCRVα 7.2 VB | Miltenyi | AB_2653669 | 1:10 |
| a-TNFα APC | Miltenyi | AB_244201 | 1:10 |
| a-TNFα eFluor450 | eBioscience | AB_2043889 | 1:20 |

Secondly, drug dosage was normalized to the lowest drug dosage per patient and drug. The lowest drug dosage at baseline was set to one, while corresponding drug dosages at other time points where either zero if the medication was discontinued, one if there was no change in drug dosage between time points, smaller than one if the drug dosage was decreased or greater than one if the drug dosage was increased at a certain time point. The sum of the agents taken was calculated at each time point.

*DNA isolation.* For DNA-based 16S rRNA gene and metagenomics sequencing, fecal samples were collected into RNALater containing tubes, shipped at room temperature and stored at −80 °C until processing. The DNA isolation protocol has been previously described[59]. Briefly, samples were treated with 500 μl of extraction buffer (200 mM Tris, 20 mM EDTA, 200 mM NaCl, pH 8.0), 200 μl of 20% SDS, 500 μl of phenol: chloroform:isoamyl alcohol (24:24:1) and 100 μl of zirconia/silica beads (0.1 mm diameter). Samples were homogenized twice with a bead beater (BioSpec) for 2 min. After precipitation of DNA, crude DNA extracts were resuspended in TE Buffer with 100 μg/ml RNase I and column purified to remove PCR inhibitors.

*16S rRNA gene amplification and sequencing.* Amplification of the V4 region (F515/R806) of the 16S rRNA gene was performed according to previously described

protocols[60,61]. Briefly, for DNA-based amplicon sequencing 25 ng of DNA was used per PCR reaction in a final volume 30 μl. The PCR conditions consisted of initial denaturation for 30 s at 98 °C, followed by 25 cycles (10 s at 98 °C, 20 s at 55 °C, and 20 s at 72 °C). Each sample was amplified in triplicates and subsequently pooled. After normalization, PCR amplicons were sequenced on MiSeq PE300 platform (Illumina) at the Helmholtz Centre for Infection Research, Braunschweig, Germany.

*Metagenomic DNA library construction and sequencing.* Sixty microliters of total DNA was used for shearing by sonication (Covaris). Fragmentation was performed as follows; processing time: 150 s, fragment size: 200 bp, intensity: 5, duty cycle: 10. Library preparation for Illumina sequencing was performed using the NEBNext Ultra DNA library prep Kit (New England Biolabs). The library preparation was performed according to the manufacturer's instructions. An input of 500 ng of sheared DNA was used and the size selection was performed using AMPure XP beads with the following parameters. First bead selection: 55 μl, and second: 25 μl. Adaptor enrichment was performed using seven cycles of PCR using NEBNext Multiplex oligonucleotides for Illumina (Set1 and Set2, New England Biolabs). Sequencing was performed on NovaSeq PE1000 platform (Illumina) at the Helmholtz Centre for Infection Research, Braunschweig, Germany.

*16S sequence processing.* Reads retrieved from 16S amplicon sequencing were analyzed using the LotuS (1.62) pipeline[62]. The pipeline includes sequence quality filtering[63], read merging[64], adapter and primer removal, chimera removal[65], clustering[66], and taxonomic classification[67] based on the SILVA (v138)[68] database. The validation dataset[22] was reprocessed using the exact same settings.

*Shotgun metagenomic processing.* Metagenomic shotgun sequences were processed within the NGLess framework (0.10)[69]. Reads were quality filtered by a minimum read length of 45 bp and a minimum Phred quality score of 25. Sequences passing that filter were mapped to the human genome (adapted from hg19; minimum 45 bp match, 90% minimum identity) and filtered. Sequences identified as non-human were mapped with bwa[70] to a) the IGC gene catalog (0.5)[70] with a minimum match size of 45 bp and a minimum identity of 95%, b) 40 reference marker genes described in Ciccarelli et al.[71] and Sorek et al.[72] with a minimum match size of 45 bp and a minimum identity of 97%. Reads mapping to the marker genes were extracted and further mapped to marker gene-based OTUs[73]. Mapping statistics can be found in Supplementary Data 14.

**Microbiome statistical analysis**
*Data pre-processing.* Reads mapped to the IGC microbial gene catalog (0.5)[71] were rarefied using the RTK (0.93.1)[74] with default settings (95% of smallest total reads —here 15,247,497 reads/sample). Reads were mapped to the mOTUv2 (2.1) taxonomic marker genes[73] were likewise rarefied (5838 reads/sample). Reads mapped to 16S OTUs (27813 reads), to ensure sample compatibility regardless of sampling depth. For functional microbiome analysis, IGC genes were binned to KEGG KOs[75] based on the annotations in MOCAT2 (2.0.1)[75], then binned by averaging over KOs to KEGG modules and to Gut Microbial Modules (GMMs)[76]. 16S and mOTUv2 (2.1) OTUs were binned at more rootwards taxonomic levels using the taxonomies provided with LotuS (1.62)[62] and the mOTUv2 (2.1) tool[73] respectively.

*Alpha and beta diversity analysis.* We assessed several metrics for gut alpha diversity using the 16S species data (thus, available in equal form for both arms), namely species richness, Shannon diversity, community evenness, Simpson's and the Inverse Simpson's metric, and the Chao1 index, calculated using the RTK (0.93.1)tool[74]. Unpaired MWU tests failed to reach significance ($P > 0.05$) for all comparisons of subsets of samples: each time point versus each other time point, in each arm separately and pooled, and between the arms within each time separately and pooled. Subsequently, we assessed within-individual changes in alpha diversity for both the DASH and the fasting+DASH arm, analogously to analysis of microbial taxa, functional modules, clinical phenotypes, and immune cell population counts, controlling for medication changes in the same manner. Supplementary Data 1 shows these results. In short, there is a nonsignificant trend for fasting to reduce diversity, which refeeding then restores, in the fasting+DASH arm, whereas no such trend is visible in the DASH-only arm. Beta diversity was assessed as community distances between samples computed using the vegan (2.5-5) R package. For microbiome data, Bray-Curtis distances on rarefied samples were used, and for immunome data, Euclidean distances. Comparisons of distance profiles was performed using Mann–Whitney U tests.

*Multivariate analysis.* Mutlivariate analysis was carried out using Principal Coordinates Analysis (PcoA) as per the vegan (2.5-5) R package, with the same distance metrics as noted above. Where described, delta metrics for the first two dimensions of unconstrained ordination were computed. PERMANOVA tests for multivariate effect were done using the adonis function in the vegan (2.5-5)[77] R package, stratified for patient ID.

*Univariate contrast analysis.* For all univariate analysis of clinical, immunome, or microbiome features, medication changes during the course of the study were accounted for as possible confounders using the following two-step procedure. The first step was a nested model comparison of a linear model for each feature, involving as predictors age, patient ID, sex, and normalized dosage of each salient medication tracked at each time point, with the same model but additionally containing time point V1-V3 as a predictor. Models were compared using a likelihood ratio test as implemented in the lmtest (0.9-37)[78] R package, and adjusted for false discovery rate (FDR) using the Benjamini–Hochberg (BH) procedure within each measurement space. In the second step, features with FDR < 0.1 were retained for a second phase of post-hoc tests using Mann–Whitney $U$ comparisons between values at each pair of time points, BH FDR-adjusted between time point comparisons ($n = 3$) and requiring FDR < 0.05 to retain the result as significant. Standardized non-parametric effect sizes were taken using the (signed) Cliff's delta metric as implemented in the orddom (3.1)[79] R package. The same methods were used to analyze the validation dataset, with the exception no drugs were adjusted for as subjects were unmedicated[22].

**Statistical analysis of 24 h ambulatory blood pressure and body weight changes.** Body weight and blood pressure change differences between Responders and Non-Responders were compared with two-sided Mann–Whitney U test using GraphPad Prism (6.01).

*Fasting arm enterotyping.* Enterotypes of the samples in the fasting arm were performed by implementing the R package DirichletMultinomial (1.32.0.)[15] on the genus-level abundance table.

*Correlation analysis.* To assess possible interactions between immune cells, taxa, and quantitative phenotypes, another two-step test was used: first a Spearman correlation test using samples pooled across time points, and with Spearman's rho used as standardized signed effect estimate. *P*-values from this were FDR-adjusted with the BH method for each comparison of two data spaces, requiring FDR < 0.05 for significance. Second, a post-hoc test was done to account for dependency between same-donor samples: for each of two correlated features, a mixed-effects model was fitted of the rank-transformed variable using the rank of the other as predictor, with patient ID as a random effect. This model was compared to a simpler model containing only the random effect under a likelihood ratio test as implemented in the lmtest (0.9–37)[78] R package. The highest P-value for the two possible such models was taken, and $P < 0.05$ was additionally required to retain the correlation as robust. Correlation was visualized by the R packages circlize[80] and pheatmap[81].

**Re analysis of previous datasets for comparison.** Samples from Kushugulova et al.[14] and Forslund et al.[11] were previously mapped to the IGC gene catalog (0.5) and the mOTU marker genes; these abundances (binned at the level of KEGG and GMM modules as per the above in case of functional profiles). The Kushugulova samples were tested for significantly differential abundances between MetS cases and controls using the Mann–Whitney U test, then controlling that a MetS status predictor still significantly improves fit (using the R lmtest ((0.9–37)[78] package) of the rank-transformed abundances when added to a linear model already incorporating metformin status as a predictor, thereby controlling for confounding influence of metformin treatment status. Analogously, the Forslund samples were tested for significantly differential abundances between metformin-treated and untreated patients using the Mann–Whitney U test, then controlling that a metformin status predictor still significantly improves fit (using the R lmtest (0.9–37)[78] package) of the rank-transformed abundances when added to a linear model already incorporating MetS status as a predictor, thereby controlling for confounding influence of MetS status. The validation dataset[22] was analyzed exactly as the main study dataset, as described above.

**Machine-learning prediction of treatment response at the single-subject level.** To estimate how well the omics data enables forecasting of the blood-pressure response in future patients, we performed a leave-one-patient-out cross-validation procedure. This approach represents the gold standard in the machine-learning community to carry out an acid-test that empirically evaluates the practical value of a predictive model[82]. To this end, the set of n participants was iteratively split into n − 1 participants as training set, and the untouched data from the hold-out participant as the test set. All input variables were z-scored by centering to zero mean and unit-scaling to a variance of one[83]. In each of n cross-validation folds, the logistic-regression algorithm was a natural choice of method for binary classification (no intercept term, L2 shrinkage penalty, hyper-parameter C defaulted to 1.0). Given that the number of variables was >10× times larger than the number of participants, dimensionality reduction was necessary for a preliminary selection of a set of ten most promising input variables that could be relevant for outcome prediction. Forward-stepwise selection is an established means[84] to screen the relevance of several hundred quantitative measures. The first step identifies the single input variable among the $p$ candidates, with the best p-value having a statistically significant association with the blood-pressure outcome. After adding this first variable to the empty null model, the second most significant (i.e., smallest p-value) was

searched based on the remaining $p − 1$ input variables. Based on a two-variable model, the third most significant variable was searched based on $p − 2$ remaining variables, and so forth. This successive identification of the ten most promising among the $p$ overall input dimensions did not bias the subsequently performed prediction performance estimate, because the entire variable reduction scheme was exclusively carried out on the $n − 1$ participants of the current cross-validation fold. Based on the top 10 variables, the logistic-regression algorithm could be more robustly fit to these subselected ten input dimensions only. The ensuing predictive model was then explicitly validated by computing whether or not the obtained model parameters allowed for accurate derivation of the relevant blood-pressure response for the independent, unseen participant. In this way, the omics data of each patient in our dataset served as test observation once. Averaging these yes-no results over all $n$ predicted, versus observed clinical responses, yielded an estimate of the expected forecasting accuracy of the predictive model in participants that we would observe in other or later acquired datasets.

**Reporting summary.** Further information on research design is available in the Nature Research Reporting Summary linked to this article.

## Data availability

Data supporting the conclusions of this manuscript will be made available by the authors, without undue reservation, to any qualified researcher. The Python code for this analysis can be found online: https://github.com/fastingproject/Fasting_Paper_2020[85]. Databases are to be found under the following links. KEGG: https://www.genome.jp/kegg/, SILVA: https://www.arb-silva.de. mOTU: https://motu-tool.org/, Mesnage dataset: https://www.ncbi.nlm.nih.gov/bioproject/PRJNA531091, IGC: https://db.cngb.org/microbiome/genecatalog/genecatalog_human/. Stool sequencing data: https://www.ncbi.nlm.nih.gov/bioproject/PRJNA698459.

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

## Acknowledgements
The authors thank Juliane Anders, Jana Czychi and Gabriele N'Diaye for the outstanding technical assistance, Falk Hildebrand, Luis Pedro Coelho, and Renato Alves for assistance with metagenomic software adaptation.

## Author contributions
A.M. led and designed and performed most experiments, analyzed and interpreted the data. N.S., H.C., G.D., A.Mi. recruited the patient and conducted the clinical study. T.S., T.R.L. performed 16S and metagenomic sequencing. A.M., H.B., E.G.A., I.H., U.S., M.K. performed immunophenotyping experiments and analyzed data. S.K.F., U.L., C.C. performed computational analyses. S.K.F., D.B., A.M. performed statistical analyses. A.M., M.K., R.D., A.M., D.N.M., S.K.F supervised the experiments and analyses. A.M., H.B., L.M., N.W., A.Ma., U.S., M.K., A.Mi., D.N.M., S.K.F. conceived parts of the project, supervised the experiments, and interpreted the data. A.M., H.B., E.G.A., D.B., A.Mi., D.N.M., and S.K.F. wrote the manuscript with key editing by L.M., R.D., M.K. and further input from all authors.

## Funding

## Competing interests
The authors declare no competing interests.

## Additional information

[1]Experimental and Clinical Research Center, a joint cooperation of Max Delbruck Center for Molecular Medicine and Charité - Universitätsmedizin Berlin, Berlin, Germany. [2]Charité - Universitätsmedizin Berlin, corporate member of Freie Universität Berlin and Humboldt-Universität zu Berlin, Berlin, Germany. [3]DZHK (German Centre for Cardiovascular Research), partner site Berlin, Berlin, Germany. [4]Max Delbruck Center for Molecular Medicine in the Helmholtz Association (MDC), Berlin, Germany. [5]Department of Biology, Chemistry, and Pharmacy, Freie Universität Berlin, Berlin, Germany. [6]Department of Internal and Integrative Medicine, Immanuel Krankenhaus Berlin, Berlin, Germany. [7]Berlin Institute of Health (BIH), Berlin, Germany. [8]Charité - Universitätsmedizin Berlin, corporate member of Freie Universität Berlin and Humboldt-Universität zu Berlin, Department of Nephrology and Internal Intensive Care Medicine, Berlin, Germany. [9]VIB Laboratory of Translational Immunomodulation, VIB Center for Inflammation Research (IRC), UHasselt, Campus Diepenbeek, Hasselt, Belgium. [10]Department of Immunology, Biomedical Research Institute, UHasselt, Campus Diepenbeek, Hasselt, Belgium. [11]Department of Internal and Integrative Medicine, Kliniken Essen-Mitte, Faculty of Medicine, University of Duisburg-Essen, Essen, Germany. [12]Department of Microbial Immune Regulation, Helmholtz Centre for Infection Research, Braunschweig, Germany. [13]Hannover Medical School, Hannover, Germany. [14]Department of Cardiology and Nephrology, HELIOS-Klinikum, Berlin, Germany. [15]Department of Biomedical Engineering, McConnell Brain Imaging Centre, Montreal Neurological Institute, Faculty of Medicine, McGill University, Montreal, Canada. [16]Mila – Quebec Artificial Intelligence Institute, Montreal, Canada. [17]Parietal Team, Institut National de Recherche en Informatique et en Automatique (INRIA), Neurospin, Commissariat à l'Energie Atomique (CEA) Saclay, Gif-sur-Yvette, France. [18]These authors contributed equally: Andreas Michalsen, Dominik N. Müller, Sofia K. Forslund. ✉email: andreas.michalsen@charite.de; dominik.mueller@mdc-berlin.de; sofia.forslund@mdc-berlin.de

