## [Peer Review File · Nature Communications]

REVIEWER COMMENTS

Reviewer #1 (Remarks to the Author):

In this manuscript, the authors describe the effects of a DASH diet alone versus fasting + DASH on metabolic outcomes, microbiome, and immune features in participants with metabolic syndrome. The authors found that fasting combined with dietary modification resulted in greater response rates and more pronounced effects than diet alone. Although many of these effects were reversible, they were along longer-lived than those achieved by diet alone. Interestingly, the authors were also able to predict response rate to the fasting + diet intervention with a ~77% success rate using baseline immune parameters.

This is a very interesting study. It demonstrates the importance of the interaction of the microbiome, immune system, and diet in shaping health outcomes, and it will likely be of broad interest to the field. That said, it would benefit from the consideration of the following:

1. Is the inclusion of 16S data necessary to communicate the outcomes of this study? Although relationships among OTUs and outcomes are presented, it appears that many of these are recapitulated with the WGS data, and the value of 280 OTUs per sample seems de minimus. Do these outcomes change if the lowest-yielding samples are dropped in favor of those that produced greater numbers of reads? Although meaningful data were produced in the era of cloning and Sanger sequencing, current methods for producing amplicon data should allow for substantially more sequence data to be included.

2. The following passage in the methods section is unclear. Did the authors mean to say that the test was performed to ensure that the signal(s) were not reproducible? "The samples were tested for significantly differential abundances between metabolic syndrome cases and controls (Kushugulova samples) or between metformin users and non-users (Forslund samples) using Mann-Whitney U tests with an additional nested model comparison post-hoc test to ensure signal (metabolic syndrome, metformin) is not reducible to the other, using a likelihood ratio test as implemented in the R lmtest40 package applied to rank-transformed data."

3. In the methods section, the full Github url should be provided (e.g., www.github.com/banilo/TO_BE_ADDED_LATER is incomplete). Please also clarify — was leave one out cross validation performed on the logistic regression after variable selection with stepwise selection? As written, this would be difficult for another to replicate.

4. Figure S1 — The use of Corona# in the figure is unclear and does not match other references to the fasting + DASH treatment group

5. pg 5, lines 6-8 -- "Neither fasting nor refeeding significantly (Mann-Whitney U (MWU) $P > 0.05$) altered microbiome species richness/alpha diversity (Shannon, Figure 1B, Supplementary Figure S1)" — The data presented in Figure S1 don't clearly support this statement. Although overall composition appears to be similar, this is not = to diversity and one does not have a sense for the degree of variation observed between subjects, treatment groups, or time points.

6. g 6 line 19 — *F. prausnitzii*, *E. rectale* and *C. comes* - This is the first instance of any of these taxa in the manuscript. Full binomials should be provided.

7. Pg 6 line 20-21 — "Interestingly, *C. comes* abundance change was predominantly driven by changes in BMI. Bacteroidaceae showed the opposite pattern and an OTU identified as an *Odoribacter* species likewise bloomed during fasting " — *C. comes* moved in concert, but causation cannot be inferred from this data.

8. Fig S5 — The links between MetS Kazakh and Metformin MetaHIT are not apparent from the figure legend. Please link these more explicitly with the referenced studies.

9. Pg 8, lines 1-6 — A review of the % of responders for the DASH arm is provided, but the same is not done for fasting + DASH.

10. Pg 12, lines 15-16: "Regarding the top ten features derived as indicative for successful patient classification, responders seem to have less of a pro-inflammatory immune signature at baseline (Figure 5C)." — This suggests that responders may have less severe disease. Could this be identified by other clinical metrics (e.g., lower end of BMI, BP, medication doses, etc)?

11. Pg 14 lines 16-18 — "In addition, some functional gut specific gene modules in our dataset were enriched only in BP-responders, for example the pyruvate:formate lyase module, MF0085. Our data are congruent with a study showing enrichment of the same enzyme in symptomatic atherosclerosis patients compared to healthy controls." — This statement seems to be incongruent. If pyruvate:formate lyase gene are enriched in BP responders, this seems to be potentially beneficial. How then, is this congruent with enrichment of the same enzyme in symptomatic atherosclerosis patients (relative to healthy controls)?

12. Table 1 — What units are represented by the +/- in this table?

Reviewer #2 (Remarks to the Author):

NCOMMS-20-05648

Fasting alters the gut microbiome with sustained blood pressure and body weight reduction in metabolic syndrome patients

Balogh et al. report about fasting that results in body weight reduction, sustained blood pressure and alterations in the gut microbiome and the gut immune system. The authors suggest that fasting exerts its beneficial effects via microbiome and immunological changes in the gut. The paper contains some interesting findings, but also some weaknesses.

Major

Lack of novelty. The authors state P3L25f. Our study is the first of its kind to investigate the effects of a lifestyle modification in combination with fasting therapy in patients with MetS using a multi-omics approach. However, a number of papers have been published on this topic with virtually similar results (e.g. Liu Z, et al. Nat Commun. 2020; Ozkul C, et al. Benef Microbes. 2020; Mesnage R, et al. J Nutr Sci. 2019; Guevara-Cruz M, et al. J Am Heart Assoc. 2019; Frost F, et al. PLoS One. 2019; Depommier C, et al., Nat Med. 2019; Ortega-Vega EL, et al. Gut Microbes. 2019; Velikonja A, et al. Anaerobe. 2019; Roager HM, et al., Gut. 2019; Kopf JC, et al. Nutr J. 2018; Louis S, et al. PLoS One. 2016).

Descriptive results. Several associations between fasting and BP or between fasting and microbiota changes or immunological changes, respectively, are described. However, underlying mechanisms and consequences are hardly presented.

P13L3: Neither the change in blood pressure nor global changes to the microbial composition or immunome appear mediated by this BMI decrease ... Which other mechanisms than BMI decrease can be anticipated?

The authors conclude that fasting is of major importance for the beneficial effects on BP and BMI, while DASH has only minor effects. Since fasting was not examined without the sub DASH diet, it cannot be excluded that fasting is effective only with the DASH diet and not by itself. This issue needs to be explored further.

P11L11: Baseline indicators predicting efficacy of fasting on blood pressure. This is of particular interest. However, the microbiota changes following fasting seem to be transient while effects on BP last at least for 3 month. This needs explanation.

By three months post intervention, propionate modules and Bacteroides sp. are almost back at baseline while BP (relative to medication dosage) remains improved, suggesting that their

The idea that transient microbiota changes during refeeding may have stabilized a less hypertensive state through downstream mechanisms is highly speculative and needs to be confirmed.

P17L16 f. The favorable impact of fasting shown here highlights this intervention as a promising non-pharmacological intervention for the treatment of MetS.

This conclusion is not justified by the data, since only BP but no other aspects of the MetS have been examined in the present study.

Although the machine learning data suggest a predictive value of microbiota patterns for outcome, such data need to be confirmed in a prospective trial prior to deduct clinical conclusions or recommendations.

Minor

Spelling, e.g. P3L13 The high-resolution multi-omics data highlight fasting as a promising non-pharmacological intervention in MetS.

REVIEWER COMMENTS

Reviewer #1 (Remarks to the Author):

In this manuscript, the authors describe the effects of a DASH diet alone versus fasting + DASH on metabolic outcomes, microbiome, and immune features in participants with metabolic syndrome. The authors found that fasting combined with dietary modification resulted in greater response rates and more pronounced effects than diet alone. Although many of these effects were reversible, they were along longer-lived than those achieved by diet alone. Interestingly, the authors were also able to predict response rate to the fasting + diet intervention with a ~77% success rate using baseline immune parameters.

This is a very interesting study. It demonstrates the importance of the interaction of the microbiome, immune system, and diet in shaping health outcomes, and it will likely be of broad interest to the field.

We thank the reviewer for this assessment of our work!

That said, it would benefit from the consideration of the following:

Q1: Is the inclusion of 16S data necessary to communicate the outcomes of this study? Although relationships among OTUs and outcomes are presented, it appears that many of these are recapitulated with the WGS data, and the value of 280 OTUs per sample seems de minimus. Do these outcomes change if the lowest-yielding samples are dropped in favor of those that produced greater numbers of reads? Although meaningful data were produced in the era of cloning and Sanger sequencing, current methods for producing amplicon data should allow for substantially more sequence data to be included.

A1: We thank the reviewer for spotting this! Indeed, we had failed to consider how only a few anomalous samples had such low coverage. By excluding them, we could rarefy to a hundredfold more reads per sample. While results largely remain the same, this allowed further refinement of the signal (including verification the Faecalibacterium result from the WGS). We consider the inclusion of the 16S data useful for two reasons: first, the internal validation of WGS signals, and second, the possibility of more direct comparison to other datasets. Reviewer #2 alerted us to an article and dataset published after we submitted, where a similar fasting protocol was applied to a healthy (as opposed to metabolically compromised) patient population. Our 16S data allows us to directly compare our results to theirs, offering further support for a number of the changes that appear to be fasting-specific, beyond the context of our study, and allowing us to test our microbiome-based predictor for fasting response in an independent dataset.

ACTION TAKEN: *In the revised manuscript, all 16S analysis was rerun using ~28K reads/sample. Results are essentially the same though a few more taxa were found to be altered significantly during the intervention, including some persisting changes at follow-up (Fig 4A, B). Comparison to a fasting 16S gut dataset in a healthy cohort was added, revealing that many of the signals generalize (Fig 8, Supplementary Table 11).*

Q2: The following passage in the methods section is unclear. Did the authors mean to say that the test was performed to ensure that the signal(s) were not reproducible? "The samples were tested for significantly differential abundances between metabolic syndrome cases and controls"

(Kushugulova samples) or between metformin users and non-users (Forsslund samples) using Mann-Whitney U tests with an additional nested model comparison post-hoc test to ensure signal (metabolic syndrome, metformin) is not reducible to the other, using a likelihood ratio test as implemented in the R lmtest40 package applied to rank-transformed data."

A2: This was indeed unclear as written, and we are thankful to the reviewer for pointing it out. To clarify: This additional analysis aimed to test what metformin signatures appear like in a dataset highly powered for that (Forsslund et al., 2015) and what metabolic syndrome signature appear like in a dataset in turn highly powered for that (Kushugulova et al., 2016). The challenge lies in that both datasets contain both metformin+/- and metabolic syndrome+/- individuals, so in addition to the primary test - is feature X differentially abundant under variable Y? - we performed a post-hoc test for the potential confounder Z. That is, in the Forsslund et al., 2015 dataset, for each feature found significantly differentially abundant in a comparison of metformin positive and negative subjects, we verified that this differential abundance remained significant when controlling for the metabolic syndrome status of the subjects. Similarly, in the Kushugulova et al., 2016 dataset, for each feature found significantly differentially abundant in a comparison of metabolic syndrome patients versus healthy controls, we verified that this differential abundance remained significant when controlling for the metformin treatment status of the subjects. Thus, all in all, the intent was to conclude a reliable, robust metabolic syndrome signature controlling for metformin treatment, and a reliable, robust metformin signature controlling for metabolic syndrome, so that we could compare our fasting and refeeding signature to either of these for assessing its role in context of either pathology or treatment.

ACTION TAKEN: *This section of the methods has now been clarified to read:*

“Samples from Kushugulova et al.⁴⁶ and Forsslund et al.⁴⁷ were previously mapped to the IGC gene catalog and the mOTU marker genes; these abundances (binned at the level of KEGG and GMM modules as per the above in case of functional profiles). The Kushugulova samples were tested for significantly differential abundances between metabolic syndrome cases and controls using the Mann-Whitney U test, then controlling that a metabolic syndrome status predictor still significantly improves fit (using the R lmtest⁴¹ package) of the rank-transformed abundances when added to a linear model already incorporating metformin status as a predictor, thereby controlling for confounding influence of metformin treatment status. Analogously, the Forsslund samples were tested for significantly differential abundances between metformin treated and untreated patients using the Mann-Whitney U test, then controlling that a metformin status predictor still significantly improves fit (using the R lmtest⁴¹ package) of the rank-transformed abundances when added to a linear model already incorporating metabolic syndrome status as a predictor, thereby controlling for confounding influence of metabolic syndrome status. The validation dataset³¹ was analyzed exactly as the main study dataset, as described above.”

Q3: In the methods section, the full Github url should be provided (e.g., www.github.com/banilo/TO_BE_ADDED_LATER is incomplete). Please also clarify — was leave one out cross validation performed on the logistic regression after variable selection with stepwise selection? As written, this would be difficult for another to replicate.

A3: *We assure the reviewer that all predictive analyses were evaluated on unseen hold-out data. That is, predictive models were trained on a majority of the data and subsequently the built*

model was carried over to yet-unseen participants and their biological data to obtain an unbiased estimate of the expected out-of-sample performance that we would observe in newly acquired data from a similar study. We are happy to supply further detail as needed.

ACTION TAKEN: Full and correct URL has been added to the source code ([https://github.com/fastingproject/Fasting Paper 2020](https://github.com/fastingproject/Fasting_Paper_2020)). The specific cross-validation scheme used (leave-one-out) is now explicitly described in the revised manuscript to allow replication.

Q4: 4. Figure S1 — The use of Corona# in the figure is unclear and does not match other references to the fasting + DASH treatment group

A4: Thank you for alerting us to this issue.

ACTION TAKEN: The Figure S1 has been corrected in this regard.

Q5: pg 5, lines 6-8 -- "Neither fasting nor refeeding significantly (Mann-Whitney U (MWU) $P > 0.05$) altered microbiome species richness/alpha diversity (Shannon, Figure 1B, Supplementary Figure S1)" — The data presented in Figure S1 don't clearly support this statement. Although overall composition appears to be similar, this is not = to diversity and one does not have a sense for the degree of variation observed between subjects, treatment groups, or time points.

A5: Point well taken! More accurate would be to say; we cannot demonstrate any significant change in any of the alpha diversity metrics we tested (which beyond the Shannon diversity shown in the figure also included species richness and the Chao1 and Simpson metrics), either in WGS or the 16S data. We cannot rule out that a larger dataset would demonstrate an effect of fasting on alpha diversity when comparing the groups, though we do not expect so.

ACTION TAKEN: We have expanded the description of the alpha diversity to read as follows:

"We could not conclude significant (between-group Mann-Whitney U (MWU) $P > 0.05$, within-individual likelihood ratio test $FDR > 0.1$ for all comparisons; Supplementary Analysis; Supplementary Table 1) changes to the microbiome species richness/alpha diversity (Shannon, Figure 1B, Supplementary Figure S1; further metrics Supplementary Analysis, Supplementary Table 1 under either fasting or refeeding in the present dataset, though a trend of reduced, then restored diversity was seen in the longitudinal tests. Similarly, we cannot conclude any significant change between time points in intersample gut taxonomic variability/beta diversity (intersample Bray-Curtis distance, Figure 1C; similar results achieved on 16S data (results not shown))."

We have added more extensive analysis (Supplementary Analysis, Supplementary Table 1) of alpha diversity differences to the supplementary materials, reflecting the questions raised by the reviewer. In particular, in terms of within-individual shifts, we now added longitudinal analysis of alpha diversity metrics using the same approach as for immune cells, microbiome taxa, clinical metrics and gut functional modules. Here a slight trend of reduced alpha diversity in fasting, restored during refeeding, can be seen in the fasting+DASH arm only, though it does not achieve significance in any comparison. Thus, our data suggests a trend in this regard similar to the other microbiome features, and we discuss this possibility in the revised manuscript.

Q6: Pg 6 line 19 — F. prausnitzii, E. rectale and C. comes - This is the first instance of any of these taxa in the manuscript. Full binomials should be provided.

A6: Thank you for alerting us to this issue.

ACTION TAKEN: The text has been extended with the full binomials as suggested.

Q7: Pg 6 line 20-21 — "Interestingly, C. comes abundance change was predominantly driven by changes in BMI. Bacteroidaceae showed the opposite pattern and an OTU identified as an Odoribacter species likewise bloomed during fasting " — C. comes moved in concert, but causation cannot be inferred from this data.

A7: This is absolutely true – while the C. comes abundance fits closer to BMI change than to the intervention itself in this particular dataset, this does mean it is driven by it, only that the data is compatible with such a scenario (whereas for most other taxa, it is not).

ACTION TAKEN: We have rephrased this sentence to read:

“Interestingly, modeling C. comes abundance change as a function of BMI change yielded a better fit of the data than when modeling it as a function of the fasting intervention.”

Q8: Fig S5 — The links between MetS Kazakh and Metformin MetaHIT are not apparent from the figure legend. Please link these more explicitly with the referenced studies.

A8: We thank the reviewer for pointing this out!

ACTION TAKEN: The legend of this figure has been rewritten as follows:

“Fig. S5. Microbiome functional potential changes during fasting are similar to those induced by metformin and unlike those characteristics for MetS. To compare the microbiome signatures of fasting and refeeding to those seen in metabolic syndrome and in the treatment of insulin resistance with metformin, we reanalyzed gut microbiome data from two previous studies powered to test these two factors, respectively. To test associations between gut microbiome functional module (GMM) abundances and metabolic syndrome status, we reanalyzed samples from Kushgulova et al.¹⁵, controlling for metformin treatment. To test associations between gut microbiome functional module (GMM) abundances and metformin treatment, we reanalyzed samples from Forslund et al.¹², controlling for metabolic syndrome status. Size and direction (Cliff’s delta) of significant effects are shown in the heatmap alongside corresponding signals from the present study. Fasting, recovery, and overall study effect from our novel data are shown. Blue and red indicate significant (MWU FDR < 0.1 for metformin/MetS status, respectively, post-hoc nested model test for confounder (the other variable) $P < 0.05$) depletion/enrichment in each data set. White indicates non-significant effect or absence of the module in a dataset. Modules significantly different in abundance in the metformin substudy show some overlap with and similar directional changes as in our fasting study, whereas recovery exhibits the opposing pattern. MetS and metformin functional signals are starkly different from one another, however there is little overlap between features altered in MetS and by fasting in our novel data.”

Q9: Pg 8, lines 1-6 — A review of the % of responders for the DASH arm is provided, but the same is not done for fasting + DASH.

A9: *Thank you for noticing this, we have extended the text with the appropriate values.*

ACTION TAKEN: *The text has been extended to provide this.*

Q10. Pg 12, lines 15-16: “Regarding the top ten features derived as indicative for successful patient classification, responders seem to have less of a pro-inflammatory immune signature at baseline (Figure 5C).” — This suggests that responders may have less severe disease. Could this be identified by other clinical metrics (e.g., lower end of BMI, BP, medication doses, etc)?

A10: *The features informing the predictor does indicate some difference in severity of pro-inflammatory immune signature between responders and non-responders. Expanding to clinical phenotype, as the reviewer suggests, we saw no significant difference in baseline blood pressure, BMI, lipid levels, or glucose homeostasis parameters between responders and non-responders. However, responders exhibited higher median systolic blood pressure than non-responders at baseline, 135 mmHg and 128 mmHg, respectively. In addition, responders had lower median BMI than non-responders; 32 and 36.5, respectively. However, body fat percentage was slightly higher in the fasting+DASH group compared to the DASH group (median 42%, 39%, respectively). Furthermore, responders had a baseline median LDL of 149 mg/dl compared to 122 mg/dl for non-responders, while HDL did not differ (in both groups 48 mg/dl). Baseline antihypertensive medication did not differ significantly between the groups (responders' normalized mean dose: 1.4, non-responders' normalized mean dose: 2.1). The data indicates, that although blood pressure responders and non-responders do demonstrate slightly different trends in their clinical parameters, responders cannot really be said to have any less severe disease.*

ACTION TAKEN: *We explicitly state this concern in the revised manuscript, and now report in detail on the clinical profiles of responders and non-responders at baseline.*

Q11: Pg 14 lines 16-18 — "In addition, some functional gut specific gene modules in our dataset were enriched only in BP-responders, for example the pyruvate:formate lyase module, MF0085. Our data are congruent with a study showing enrichment of the same enzyme in symptomatic atherosclerosis patients compared to healthy controls." — This statement seems to be incongruent. If pyruvate:formate lyase gene are enriched in BP responders, this seems to be potentially beneficial. How then, is this congruent with enrichment of the same enzyme in symptomatic atherosclerosis patients (relative to healthy controls)?

A11: *We apologize for any confusion - this was a poorly worded statement on our part. This module has higher abundance in responders than non-responders at baseline (normalized Cliff's delta effect size 0.33) though this does not achieve significance. It then decreases in abundance during refeeding, achieving significance only within the blood pressure-responder group. Thus, higher levels pre-intervention appears rather to be a marker of poorer health, as supported by its enrichment in atherosclerosis, and its reduction during refeeding indicative of normalization towards eubiosis.*

ACTION TAKEN: *The corresponding paragraph has been modified in text for clarity.*

“In addition, abundance of some gut functional gut-specific gene modules were significantly altered in our dataset only in BP-responders, for example of the

pyruvate:formate lyase module, MF0085, which was decreased after refeeding. This decrease (from a trending elevation at baseline) may contribute to vascular health, as a recent study demonstrated enrichment of the same enzyme in atherosclerosis patients relative to healthy controls³¹, and formate production has been previously linked to blood pressure regulation^{32,33}.”

Q12: Table 1 - What units are represented by the +/- in this table?

A12: The +/- signs denote +/- one standard deviation.

ACTION TAKEN: *The table and the legend has been modified to clarify.*

Reviewer #2 (Remarks to the Author):

NCOMMS-20-05648

Fasting alters the gut microbiome with sustained blood pressure and body weight reduction in metabolic syndrome patients

Balogh et al. report about fasting that results in body weight reduction, sustained blood pressure and alterations in the gut microbiome and the gut immune system. The authors suggest that fasting exerts its beneficial effects via microbiome and immunological changes in the gut. The paper contains some interesting findings, but also some weaknesses.

We thank the reviewer for this feedback, which allowed us to substantially strengthen our work!

Major

Q1: Lack of novelty. The authors state P3L25f. Our study is the first of its kind to investigate the effects of a lifestyle modification in combination with fasting therapy in patients with MetS using a multi-omics approach. However, a number of papers have been published on this topic with virtually similar results (e.g. Liu Z, et al. Nat Commun. 2020; Ozkul C, et al. Benef Microbes. 2020; Mesnage R, et al. J Nutr Sci. 2019; Guevara-Cruz M, et al. J Am Heart Assoc. 2019; Frost F, et al. PLoS One. 2019; Depommier C, et al., Nat Med. 2019; Ortega-Vega EL, et al. Gut Microbes. 2019; Velikonja A, et al. Anaerobe. 2019; Roager HM, et al., Gut. 2019; Kopf JC, et al. Nutr J. 2018; Louis S, et al. PLoS One. 2016).

A1: We deeply thank the reviewer for this useful feedback, especially further guiding our attention to this literature, which has substantially improved our work! Of these, some were published after submission of our article (there were unfortunately delays in the review process). Nevertheless, we have revised the paper to engage in detail with this literature as outlined below. However, none of the cited studies combines both high-resolution microbiome and immune profiling in a metabolic syndrome human intervention cohort, and as such have not been able to demonstrate, as we do, improvement in a human setting of this clinical condition and detail the accompanying microbiome and immune state changes. Furthermore, we show that the blood pressure outcome can be predicted to an extent from individual baseline data. Each of these suggested works has added crucial pieces of the puzzle that helps us put our results into context, and we have thus extended our discussion of them.

ACTIONS TAKEN: *i) We have corrected the claim of novelty to be substantially more precise rather than overreaching, as per the above. ii) Further, we cite now additional literature in the*

main text compiling prior knowledge on fasting interventions in this manner. iii) More interesting still, one study (Mesnage et al., 2020) constitutes a highly complementary intervention using a roughly similar protocol, except in a healthy rather than metabolically compromised cohort. While their cohort is smaller (n=15 in total) and only contains male subjects, with no equivalent to our immune profile data, we acquired the clinical and amplicon sequence data from the study authors and reanalyzed it jointly with our own data (Supplementary Table 11). iv) Additionally, for the first time we enterotyped our fasting+DASH dataset revealing trends consistent with the alpha and beta diversity results (Supplementary Figure S7), and through improvements to the 16S analysis approach suggested by Reviewer #1, now evidence even clearer a robust microbiome signature associated with blood pressure decrease. Thus, we can report in the revised manuscript how 1) several of our fasting-associated microbiome changes generalize to a healthy population and a different setting, and, 2) using microbiome data at baseline only, we can predict blood pressure decrease upon fasting not only in our own cohort, but in the Mesnage cohort in healthy volunteers, which was not used to build the model. Inclusion of this literature and reanalysis of compatible data for our endpoints let us validate some of our findings.

Q2: Descriptive results. Several associations between fasting and BP or between fasting and microbiota changes or immunological changes, respectively, are described. However, underlying mechanisms and consequences are hardly presented.

A2: It is true that showing full evidence of mechanism will require additional experimental studies. We plan such work, including assessing the bacteria most strongly associated with blood pressure reduction in vivo under comparative colonization of gnotobiotic mice challenged with angiotensin II. In addition, we plan a murine fasting study, although periodic fasting protocols in rodents impose several restrictions limiting a direct comparison to the human setting. Further, due to COVID-19 containment restrictions and acquiring ethics approval for such rodent studies, carrying out these experiments may take more than a year and is therefore beyond the scope of the present human study. In the meantime, through the feedback of both reviewers, in our revised manuscript we are now able to more clearly outline the mechanisms we think underlie the efficacy of the intervention, which is described in detail below.

ACTIONS TAKEN: *Through comparative analysis of a second fasting microbiome dataset (Mesnage et al.), we now show that several microbiome changes upon fasting appear generalizable and thus likely caused by the intervention rather than being idiosyncrasies of our dataset, as well as demonstrating generalizability of our treatment responder predictor. Moreover, we have revised 16S amplicon analysis based on feedback from Reviewer #1 to make use of substantially more of the available microbiome sequence material. This allowed us to highlight more clearly how favorable blood pressure response in metabolically compromised subjects is associated with normalization of a pre-intervention depletion of core short-chain fatty acid producers such as *Faecalibacterium*, which is mechanistically credible given previous literature e.g. on anti-inflammatory effects. We now clearly highlight these throughout the revised manuscript, while recognizing the limits of a descriptive analysis.*

Q3: P13L3: Neither the change in blood pressure nor global changes to the microbial composition or immunome appear mediated by this BMI decrease ... Which other mechanisms than BMI decrease can be anticipated?

A3: This is an interesting question! Our data (Fig. 2) clearly show that the decrease in systolic blood pressure cannot be explained solely by a decrease in weight or BMI. Our chord plot for blood pressure (Fig. 5) and for BMI (Extended Data Fig. 7) demonstrate distinct associations between immune cell subsets and microbial taxa, respectively, which corroborates the interpretation of the reviewer. Nevertheless, we show that a 5-day fast with less than 350 kcal per day exerted an effect on microbiome composition and immune cell subsets. Our data (comparing V1 vs. V2) suggest that microbiome and immune cells may reset to some extent during and after the intense caloric restriction during fasting, similar to a preconditioning mechanism causing the subsequent DASH diet to act differently depending on whether this precondition took place or not. This interpretation is supported by our data demonstrating that DASH diet alone neither reduced systolic blood pressure nor BMI, while affecting completely different (and substantially fewer) immune cell subsets. In line with the preconditioning hypothesis, we consider that 1) those subjects who most benefit with regards to blood pressure by a fasting+DASH intervention are those depleted at baseline for both SCFA producing taxa and SCFA production gene modules; 2) that such taxa and gene modules enrich either during the fasting phase or the refeeding phase thus ameliorating the baseline depletion mentioned in the previous point; 3) that at least some enrichment remains at three month follow-up in blood pressure-responders (and less so in non-responders). Our interpretation is that one crucial mechanism for the improvement stems from the effects of increased SCFA availability, either locally in the intestine (impacting immune signaling and intestinal permeability), systemically, or both.

ACTION TAKEN: We elaborate on the above scenario, as well as other possibilities, in the revised manuscript.

Q4: The authors conclude that fasting is of major importance for the beneficial effects on BP and BMI, while DASH has only minor effects. Since fasting was not examined without the sub DASH diet, it cannot be excluded that fasting is effective only with the DASH diet and not by itself. This issue needs to be explored further.

A4: We agree that in our setup, it is not possible to separate the effect of fasting alone from fasting followed by (and synergistic with) a DASH diet. We can only claim fasting is required for the DASH diet to achieve this effect in our cohort. Although, some effects are replicated in the similar dataset from healthy humans (without metabolic syndrome and without DASH intervention) the reviewer has referred us to (Mesnage et al.), thus indicating the precise DASH setup may not be strictly needed. Our view is that most likely, the two components of the intervention indeed synergize - fasting likely potentiates the microbiome in these patients to be shifted to a more DASH-compatible microbiota upon diet change. DASH, which is rich in fibers, can furthermore “fuel” the beneficial microbiome and can additionally contribute to cardiovascular health, which may play a part in the extended maintenance of this microbial state.

ACTION TAKEN: We have made the above limitation (and interpretation of likely synergy) clearer in the revised manuscript.

Q5: P11L11: Baseline indicators predicting efficacy of fasting on blood pressure. This is of particular interest. However, the microbiota changes following fasting seem to be transient while effects on BP last at least for 3 month. This needs explanation.

A5: The reviewer is correct that the majority of changes are (at least partially) reversible by the three month mark. Nevertheless, some microbial changes persist over time and are blood

pressure responder-specific. These sustained changes are now emphasized in Fig. 4A. We agree that the baseline indicators predicting efficacy of fasting on blood pressure are of high interest. Using the Mesnage et al. 16S and blood pressure data, we tested whether our identified indicators also could predict blood pressure responsiveness in that independent cohort. Although the Mesnage dataset consists only of n=15 healthy men, we were able to predict blood pressure responsiveness with a machine learning model trained on our MetS data, achieving 67% accuracy for blood pressure response in the Mesnage et al. subjects, which we consider rather impressive. These data are now included as Fig. 8C.

ACTION TAKEN: We have restructured Fig. 4 to highlight Fig. 4A taxa, which demonstrate a persistent study effect (followup vs baseline; V1 vs V3). We also expanded our response prediction approach as shown in Fig. 8, including the aforementioned validation analyses in an independent dataset in Fig. 8C. These results are discussed in detail in the revised manuscript.

Q6: The idea that transient microbiota changes during refeeding may have stabilized a less hypertensive state through downstream mechanisms is highly speculative and needs to be confirmed.

A6: Indeed, the core question is, how can a transient intervention and shift result in persistent health improvement? We consider this remarkable observation of major importance, especially as it is partially reproducible in the Mesnage et al. cohort that this reviewer alerted us to. Further verification of mechanism will be a major future effort for us. However at the moment with the increased resolution available based on the feedback from Reviewer #1 and making fuller use of the amplicon data, we do identify several gut microbiome features, including a core SCFA producer, which do not return fully to baseline in our blood pressure responders, consistent with the overall mechanism we propose. We strongly agree that our proposed explanation (please also see A3) is only a hypothesis suggested by the data at this stage, and underscore in the revised manuscript the need for future studies to verify it in more detail.

ACTIONS TAKEN: We have revised the manuscript to highlight the possible explanations for the observed persistent clinical outcome and transient effects on the microbiome and immunome. We also outline the further studies which would help to clarify these results. Furthermore, we have now included extended and revised analysis which details those changes to the in microbiome which do persist even at follow-up, and the possible ways in which these may also be drivers of the therapeutic impact of our intervention. Last, we have revised the discussion to acknowledge, as the reviewer notes, that this is still only a hypothesis consistent with the data we have generated, and that as such it will require further follow-up in order to be robust evidence.

Q7: P17 L16 f. The favorable impact of fasting shown here highlights this intervention as a promising non-pharmacological intervention for the treatment of MetS. This conclusion is not justified by the data, since only BP but no other aspects of the MetS have been examined in the present study.

A7: We fully agree with the reviewer and recognize our wording was not precise.

ACTION TAKEN: We have revised the manuscript throughout to clarify that the demonstrated efficacy applies to the hypertension specifically, avoiding such claims for other components of the metabolic syndrome. We revised the aforementioned noted paragraph to read:

“The favorable impact of fasting followed by DASH refeeding phase shown here highlights this intervention as a promising non-pharmacological intervention for the treatment of high blood pressure in MetS patients.”

Q8: Although the machine learning data suggest a predictive value of microbiota patterns for outcome, such data need to be confirmed in a prospective trial prior to deduct clinical conclusions or recommendations.

A8: We agree completely with the reviewer in this regard.

***ACTIONS TAKEN:** First, we now explicitly state this limitation throughout the manuscript to prevent readers from misunderstanding the implications of our results. Second, thanks to the reviewer for alerting us to the Mesnage et al. paper, which we used to validate, with some minor limitations, the predictive value of baseline microbiota for blood pressure reduction resultant from fasting in an unrelated, largely healthy cohort. The revised manuscript highlights this finding (Fig. 8C), but also states clearly that further prospective trials are needed.*

Minor

Q9: Spelling, e.g. P3L13 The high-resolution multi-omics data highlight fasting as a 14 promising non-pharmacological intervention in MetS.

A9: Thank you for alerting us to this issue!

***ACTION TAKEN:** The entire manuscript has now been proof-read by a native English-speaking author to improve language.*

REVIEWER COMMENTS

Reviewer #1 (Remarks to the Author):

With this revised submission, the authors have addressed each of the concerns identified previously. The passages that were previously ambiguous are clearer now, and missing links and/or details related to the analysis have been added.

Reviewer #2 (Remarks to the Author):

Reviewer 2

The revised paper has improved; however, some major issues remain unsolved. In particular, some of the discussions of the point-to-point reply need to be included in the Discussion. Moreover, changes and additions need to be better indicated with page and line numbers.

Q1/A1: The authors now cite the recommended literature, but they do not discuss it in Discussion. Just stating that There is a growing interest in understanding how dietary interventions shape the gut microbiome and interact with metabolic diseases, including obesity, MetS, type 2 diabetes, and (cardiovascular) health^{8, 9, 10, 24, 25, 26, 27, 28}. is not sufficient. A detailed comparison what these studies (except Mesnage et al.) showed and what was added by the present study is mandatory.

Q2/A2: Yes, full evidence of mechanism will require additional experimental studies, but a detailed discussion of possible or even probable evidences would be helpful also. These mandatory additions to the Discussion should be clearly indicated with page and line numbers in the revised manuscript.

Q3/A3: The answer in the point-to point reply is good; , but it is not included in the Discussion. To avoid the discussion becoming much longer, many passages that basically only contain a repetition of the results should be shortened significantly.

Q4/A4: Also this issue that is now addressed in the point-to-point reply needs to be included in the Discussion and clearly indicated with page and line numbers in the revised manuscript.

REVIEWER COMMENTS

Q1: The authors now cite the recommended literature, but they do not discuss it in Discussion. Just stating that There is a growing interest in understanding how dietary interventions shape the gut microbiome and interact with metabolic diseases, including obesity, MetS, type 2 diabetes, and (cardiovascular) health^{8, 9, 10, 24, 25, 26, 27, 28} is not sufficient. A detailed comparison what these studies (except Mesnage et al.) showed and what was added by the present study is mandatory.

A1: We followed the advice of the reviewer and now provide more details in the Discussion as listed below. We also added a new table (**Supplementary Table S14**), which allowed us to provide more details on each respective study without making the main text discussion too lengthy.

Exact modifications were done:

P15 L4-12

P17 L21-25

P20 L21- P21 L2

Supplementary Table S14

Q2: Yes, full evidence of mechanism will require additional experimental studies, but a detailed discussion of possible or even probable evidences would be helpful also. These mandatory additions to the Discussion should be clearly indicated with page and line numbers in the revised manuscript.

A2: As suggested by the reviewer, we have further discussed the finding and have here highlighted text blocks which were added during the previous revision as well as new text segments added now. The corresponding sections are highlighted. Please see details below.

P17 L17-19

P15 L24-P16 L16

P21 L23- P22 L4

P22 L2-4

Q3: The answer in the point-to point reply is good, but it is not included in the Discussion. To avoid the discussion becoming much longer, many passages that basically only contain a repetition of the results should be shortened significantly.

A3: As suggested by the reviewer, we now incorporated this section into the discussion.

P15 L24-P16 L16

We also made attempts to shorten the discussion where possible, though much of the description of the data was left in to avoid impairing the logic flow (indicated by the cross out in red).

P17 L17-19

P19 L11-17

P21 L5-L6

Q4: Also this issue that is now addressed in the point-to-point reply needs to be included in the Discussion and clearly indicated with page and line numbers in the revised manuscript.

A4: As suggested by the reviewer, we have added the respective section (which partially addresses Q2) in the discussion on **P21 L23- P22 L4**.